# Particle dynamics on torsional galilean spacetimes

José Figueroa-O'Farrill[1*], Can Görmez[2†] and Dieter Van den Bleeken[2,3‡]

**1** School of Mathematics and Maxwell Institute, The University of Edinburgh,
Edinburgh EH9 3FD, Scotland, UK
**2** Physics Department, Boğaziçi University, 34342 Bebek Istanbul, Turkey
**3** Institute for Theoretical Physics, KU Leuven, 3001 Leuven, Belgium

\* j.m.figueroa@ed.ac.uk , † can.gormez@boun.edu.tr ,
‡ dieter.van@boun.edu.tr

## Abstract

We study free particle motion on homogeneous kinematical spacetimes of galilean type. The three well-known cases of Galilei and (A)dS–Galilei spacetimes are included in our analysis, but our focus will be on the previously unexplored torsional galilean spacetimes. We show how in well-chosen coordinates free particle motion becomes equivalent to the dynamics of a damped harmonic oscillator, with the damping set by the torsion. The realization of the kinematical symmetry algebra in terms of conserved charges is subtle and comes with some interesting surprises, such as a homothetic version of hamiltonian vector fields and a corresponding generalization of the Poisson bracket. We show that the Bargmann extension is universal to all galilean kinematical symmetries, but also that it is no longer central for nonzero torsion. We also present a geometric interpretation of this fact through the Eisenhart lift of the dynamics.

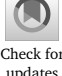

# 1   Introduction

Colloquially and in a physics context, a kinematical Lie algebra is a Lie algebra containing time and space translations, rotations and boosts, with the assumption that boosts and space translations are vectors under rotations, while time translations are invariant under rotations. Examples of kinematical Lie algebras are the isometry Lie algebras of the maximally symmetric lorentzian manifolds (Minkowski and (anti) de Sitter spacetimes), the Galilei and Carroll algebras, the Newton–Hooke algebras, as well as a host of other anonymous Lie algebras which have been known since the pioneering work of Bacry and Lévy-Leblond [1,2].

The Lie groups corresponding to these kinematical Lie algebras act transitively on so-called kinematical spacetimes, which have recently been classified [3]. Among them we find, of course, Minkowski and (anti) de Sitter spacetimes, but also their nonrelativistic limits: Galilei and (anti) de Sitter–Galilei spacetimes, as well as their ultra-relativistic limits: Carroll and (anti) de Sitter–Carroll spacetimes. There are many more, but they all fall into families: lorentzian, galilean, carrollian and aristotelian, depending on the invariant geometric structure that they possess.

It is one of the postulates of general relativity that free particle motion in a lorentzian spacetime (such as Minkowski or (anti) de Sitter) follows (causal) geodesics of the Levi-Civita connection, and it is therefore a natural question to ask about free particle motion in the other non-lorentzian kinematical spacetimes.

For example, the dynamics of a standard nonrelativistic free particle, with Lagrangian $L = \frac{1}{2}\dot{x}^a\dot{x}^a$, is well-known to be invariant under the Galilei group. This can be explained by the fact that the extrema of that Lagrangian can be interpreted as geodesics of the invariant connection on Galilei spacetime which is compatible with the invariant galilean structure. Since this spacetime is a homogeneous spacetime for the Galilei group there is a natural action of this group and because the connection is invariant, the group acts as symmetries of geodesic motion.

With one exception — namely, the carrollian lightcone — all kinematical spacetimes have invariant connections [3,4] and hence we may define free particle motion on a kinematical spacetime as geodesics relative to any invariant compatible connection.

In this paper we shall be concerned only with galilean kinematical spacetimes. In general spacetime dimension (here $\geq 4$), there are three symmetric spatially isotropic homogeneous galilean spacetimes: Galilei and (anti) de Sitter–Galilei and two one-parameter families of

torsional galilean spacetimes with a common galilean limit. As mentioned above, free particle motion on Galilei spacetime is understood. Similarly, one can study free particle motion on (anti) de Sitter–Galilei spacetime, either intrinsically or by taking the nonrelativistic limit of geodesic motion on (anti) de Sitter spacetime. These were studied in [5], who called them Newton–Hooke spacetimes, and showed that the effect of the cosmological constant $\Lambda$ was to modify the standard nonrelativistic free particle Lagrangian by the addition of a potential $L = \frac{1}{2}\dot{x}^a\dot{x}^a + \frac{1}{2}\Lambda x^a x^a$, such that force is restorative for $\Lambda < 0$ (anti de Sitter) and repulsive for $\Lambda > 0$ (de Sitter). Motion on (anti) de Sitter–Galilei spacetime has recently been considered in [6] in the context of AdS/CFT, see also [7] for an earlier discussion of the invariant wave equation.

Particle motion in the torsional galilean spaces had remained unexplored until now and this was the main motivation for this work. As we will see, the effect of the torsion is simply to add a dampening force to the motion. The damped harmonic oscillator has a long history, reviewed, e.g., in [8]. Our novel perspective – to interpret it as motion invariant under a torsional galilean kinematical symmetry algebra – reveals some relations among the conserved charges that so far went unnoticed. Although the galilean kinematical symmetries act on phase space as usual, i.e., as a subgroup of diffeomorphisms, the time translation is neither hamiltonian nor symplectic when the torsion is non-zero. Since it remains homothetically hamiltonian one can still associate a conserved charge, but the Poisson algebra of conserved charges is no longer homomorphic to the algebra of vector fields generating the kinematical symmetries. We show however that there exists a natural Lie bracket on the space of phase space functions extended with a scaling weight, that makes this space homomorphic as a Lie algebra to the algebra of homothetic hamiltonian vector fields. Under this bracket the conserved charges do form an algebra homomorphic to the torsional galilean symmetry algebra. More precisely the algebra of conserved charges is a one-dimensional extension, well known in the case without torsion as the Bargmann extension [9], of the kinematical algebra. When the torsion is non-zero this Bargmann extension is no longer central. We proceed to show that, as in the case without torsion [10], this extension can be given a geometric interpretation through the Eisenhart lift [11].

The results in this paper lead to a few natural questions that could be interesting to investigate further. How does the galilean kinematical Lie algebra embed into the full symmetry group of the damped harmonic oscillator [12]? What is the rôle of the torsional kinematical symmetry and its (non-central) Bargmann extension in the quantum theory? Is the notion of free particle defined via geodesics of invariant connections in a non-lorentzian geometry equivalent to the notion of elementary particle in the sense of Souriau [13, 14]?

This paper is organized as follows. In Section 2 we review the geometry of the torsional homogeneous galilean spacetimes, deriving some coordinate expressions and paying particular attention to the invariant connections. In particular, in Section 2.1 we define the spacetimes in terms of their Klein pairs, in Section 2.2 we introduce modified exponential coordinates which we prove to be global coordinates and in Section 2.3 we give explicit expressions for the Christoffel symbols of the invariant connections in these coordinates. In Section 3 we discuss geodesic motion on the galilean spacetimes relative to the invariant connection. More concretely, in Section 3.1 we show that in a convenient parametrization, the geodesic equation relative to any invariant connection reduces to that of a damped harmonic oscillator. Then in Section 3.2, we discuss the realization of the kinematical Lie algebra as symmetries of this equation and in terms of conserved charges. In Section 4 we re-examine geodesic motion in terms of motion in a Newton–Cartan geometry and we relate it, via the Eisenhart lift, to null geodesic motion on certain homogeneous pp-waves. The paper ends with four appendices. In Appendix A we record some definitions about Type I Newton–Cartan geometry, particularly the notion of compatible NC-doublets and NC-triplets and their use in defining an affine con-

nection compatible with the Newton–Cartan structure. In Appendix B we discuss the notion of symplectic and hamiltonian homotheties, which play an important rôle in Section 3.2. In particular we motivate and define a modified bracket on phase space functions, which we believe is new and might find applications outside the context of this paper as well. In Appendix C we extend the discussion of the main text to galilean spacetimes in spacetime dimension $\leq 3$, where particularly in dimension 3 there is a richer family of homogeneous galilean spacetimes. Finally, in Appendix D we comment on the freedom to conformally redefine the Eisenhart lift and use this to connect some of our results to those of [15].

## 2 Kinematical homogeneous spacetimes of galilean type

In this section we review the kinematical algebras of galilean type, the associated homogeneous spacetimes and the key invariant geometric structure – the invariant NC-compatible connection – that differentiates them. With the aim of keeping the discussion brief and as concrete as possible, we only summarize those results of relevance to the remainder of the paper and do so in a language and notation adapted to the problem discussed there. We refer to [3,4] for a more general discussion of kinematical homogeneous spaces of all types, as well as a more precise and detailed discussion of the galilean case.

### 2.1 Kinematical Lie algebras of galilean type

Our starting point is the Lie algebra of (infinitesimal) symmetries in $d$ spatial dimensions, a generalization of the well-known galilean algebra of a nonrelativistic free particle.

A *kinematical Lie algebra of galilean type* is a real Lie algebra for which there exists a basis $\{J_{ab} = -J_{ba}, B_a, H, P_a\}$, $a, b = 1, \ldots, d$ with Lie brackets

$$
\begin{aligned}
[J_{ab}, J_{cd}] &= \delta_{bc} J_{ad} - \delta_{ac} J_{bd} - \delta_{bd} J_{ac} + \delta_{ad} J_{bc}\,, \\
[J_{ab}, B_c] &= \delta_{bc} B_a - \delta_{ac} B_b\,, \\
[J_{ab}, P_c] &= \delta_{bc} P_a - \delta_{ac} P_b\,, \\
[J_{ab}, H] &= 0\,,
\end{aligned}
\tag{2.1}
$$

and[1]

$$
[H, B_a] = -P_a\,, \qquad [H, P_a] = \alpha B_a + \beta P_a\,, \qquad [P_a, B_b] = [P_a, P_b] = [B_a, B_b] = 0\,, \tag{2.2}
$$

where $\alpha, \beta$ are arbitrary real constants. We refer to this Lie algebra as $\mathfrak{k}_{(\alpha,\beta)}$.

The first set of brackets (2.1) is universal to all kinematical lie algebras, by their very definition. The $J_{ab}$ generate a rotation subalgebra under which the generators $B_a$ and $P_a$ transform as vectors, while $H$ is a scalar. A basis independent definition of a *kinematical Lie algebra* is a Lie algebra that contains an $\mathfrak{so}(d)$ subalgebra with respect to which the whole algebra decomposes as $\mathfrak{so}(d) \oplus 2V \oplus S$ where $2V$ are two copies of the $d$-dimensional (vector) irreducible representation of $\mathfrak{so}(d)$ and $S$ is the one-dimensional (scalar) trivial representation of $\mathfrak{so}(d)$. Some well known examples other than the galilean ones we restrict attention to in this paper, are the Poincaré and Carroll algebras.

The Lie brackets between the generators $\{H, B_a, P_a\}$ are left free by the definition of kinematical Lie algebra and are only constrained by the Jacobi identity. The brackets (2.2), which are only a subset of the possible solutions to the Jacobi identities, then select those kinematical

---

[1]Our definition here and our further discussion in the main text is only complete when $d \geq 3$. When $d = 2$ there exists an additional kinematical algebra of galilean type where the brackets (2.2) are slightly modified. See Appendix C for the discussion when $d \leq 2$.

algebras that we call 'of galilean type'.[2] Indeed, the class of algebras $\mathfrak{k}_{(\alpha,\beta)}$ defined by (2.1) and (2.2), contains the Galilei algebra $\mathfrak{k}_{(0,0)}$ as well as the galilean (A)dS – or Newton–Hooke – algebras $\mathfrak{k}_{(\pm 1,0)}$. Less familiar algebras can be obtained by choosing $\beta \neq 0$ and these will be the main focus of this paper. There is some redundancy in the parameters $(\alpha,\beta)$, as well as through mixing of the $P_a$ and $B_a$: different choices can lead to isomorphic Lie algebras. We will discuss this in more detail in the next subsection at the level of the associated homogeneous spacetimes.

Finally let us mention that based on the above, one can define a kinematical Lie group of galilean type simply to be a Lie group whose Lie algebra is kinematical of galilean type. We'll refer to the (simply connected) Lie group with Lie algebra $\mathfrak{k}_{(\alpha,\beta)}$ as $\mathcal{K}_{(\alpha,\beta)}$.

## 2.2 Homogeneous spacetimes and modified exponential coordinates

As a physicist, one would intuitively interpret the generators of the kinematical algebra — (2.1) and (2.2) — as spatial rotations $J_{ab}$, time and space translations $H$, $P_a$ as well as boosts $B_a$. One should be aware however that such interpretation is associated to an action of the associated transformations on a spacetime. It is only this action which distinguishes the boosts $B_a$ from the translations $P_a$: while boosts leave the origin invariant the translations do not. If we assume the group action to be transitive then the spacetime will be a homogeneous space that can be identified with a coset space $\mathcal{K}/\mathcal{H}$ of the kinematical Lie group $\mathcal{K}$.

The mathematical formulation of our physical intuition is then that a (homogeneous) kinematical spacetime corresponds[3] to a *Klein pair* $(\mathfrak{k}, \mathfrak{h})$. Such that $\mathfrak{k}, \mathfrak{h}$ are the Lie algebras of $\mathcal{K}$ and its subgroup $\mathcal{H}$ respectively, and we require $\mathfrak{h}$ to contain precisely the rotations $\mathfrak{so}(d)$ and one vector representation $V$, i.e.

$$\mathfrak{h} = \mathfrak{so}(d) \oplus V \subset \mathfrak{k} = \mathfrak{so}(d) \oplus 2V \oplus S\,. \tag{2.3}$$

This last condition imposes that rotations and one of its vector representations, the boosts, leave spacetime points invariant (i.e. generate the stabilizer subgroup). The complement $V \oplus S$ are the space translations (a vector) and the time translation (a scalar). It is the choice of $\mathfrak{h}$ which distinguishes the boosts from translations inside the subspace $2V$ of $\mathfrak{k}$.

Since in this paper we will always have a homogeneous kinematical spacetime – i.e. a pair $(\mathfrak{k}, \mathfrak{h})$ – in mind, we'll simply indicate the choice of $\mathfrak{h}$ through our notation for the basis $\{J_{ab}, B_a, P_a, H\}$ of $\mathfrak{k}$. From now on we'll assume that this notation implies that $\{J_{ab}, B_a\}$ is a basis of $\mathfrak{h}$. In other words, with this additional interpretation of the notation, the choice of basis (2.1, 2.2) for a kinematical Lie algebra of galilean type directly defines a kinematical

---

[2]For a classification of (spatially isotropic) kinematical Lie algebras, see [1,2,16] (for $d = 3$), [17] (for $d > 3$) and [18] (for $d = 2$). For $d = 1$ every three-dimensional Lie algebra is kinematical, so the classification goes back to Bianchi [19,20]. Note that in the literature the 'type' nomenclature is often reserved for Klein pairs $(\mathfrak{k}, \mathfrak{h})$ of a kinematical Lie algebra $\mathfrak{k}$ together with a subalgebra $\mathfrak{h}$, see, e.g., [3], where Klein pairs are referred to as Lie pairs. These pairs fall into the distinct classes of lorentzian-, riemannian-, galilean-, carrollian-, aristotelian- and low-dimensional exotic- type. One can extend this typification of the pairs to the kinematical algebras themselves, by defining such an algebra to be of a particular type if it allows a Klein pair of that type. The only subtlety is that with this definition a kinematical Lie algebra can be of more than one type. For example, the Poincaré algebra is of both lorentzian and carrollian type, as illustrated in the example on page 9 of [3] or more recently also in [21], which displays a number of homogeneous spaces of the Poincaré group of different types describing the asymptotic geometry of Minkowski spacetime. Fortunately, there is no such ambiguity for the galilean type: only galilean Lie algebras admit (spatially isotropic) galilean Klein pairs.

[3]More formally a *homogeneous kinematical spacetime* is a connected smooth manifold $\mathcal{M}$ with a transitive (and locally effective) action by a kinematical Lie group $\mathcal{K}$, whose typical stabilizer subgroup $\mathcal{H}$ has a Lie algebra $\mathfrak{h}$ which is given by $\mathfrak{so}(d) \oplus V$ as an $\mathfrak{so}(d)$ representation. It follows from this definition that any homogeneous kinematical space-time $\mathcal{M}$ is ($\mathcal{K}$-equivariantly) diffeomorphic to the coset space $\mathcal{K}/\mathcal{H}$. This then allows to identify it with the pair $(\mathfrak{k}, \mathfrak{h})$ given a number of further technical conditions, such as the pair being *effective* and *geometrically realizable*. Full details can be found in e.g. [3].

homogeneous spacetime of galilean type. Since $(\alpha, \beta)$ are the only free parameters in (2.1, 2.2), which furthermore do not appear in the brackets of $\mathfrak{h}$, we can indicate the corresponding homogeneous spacetime as

$$\mathcal{M}_{(\alpha,\beta)} = \mathcal{K}_{(\alpha,\beta)}/\mathcal{H}. \tag{2.4}$$

Let us now define global coordinates on $\mathcal{M}_{(\alpha,\beta)}$, that we will refer to as *modified exponential coordinates*. We may choose the coset of the identity element $o = e\mathcal{H} \in \mathcal{M}_{(\alpha,\beta)}$ as our origin. A point $p \in \mathcal{M}_{(\alpha,\beta)}$ is then given the coordinates $(t, x^a)$ via the definition

$$p = e^{tH} e^{x^a P_a} \cdot o. \tag{2.5}$$

This defines a smooth map $j : \mathbb{R}^{d+1} \to \mathcal{M}_{(\alpha,\beta)}$ sending $(t, x^a) \mapsto e^{tH} e^{x^a P_a} \cdot o$. This map is a local diffeomorphism. Indeed, we can see that its derivative has full rank by simply computing the pull-back of the left-invariant Maurer–Cartan one-form on $\mathcal{K}_{(\alpha,\beta)}$ via the map $\sigma : \mathbb{R}^{d+1} \to \mathcal{K}_{(\alpha,\beta)}$ sending $(t, x^a) \mapsto e^{tH} e^{x^a P_a}$. Doing so we find that

$$\sigma^{-1} d\sigma = H dt + (dx^a + \beta x^a dt)P_a + \alpha x^a dt B_a, \tag{2.6}$$

from where we read off the coframe $\theta = (dt, dx^a + \beta x^a dt)$, which is clearly everywhere invertible. Next we observe that $\mathcal{M}_{(\alpha,\beta)}$ is acted on transitively by the solvable subgroup $\mathcal{B}_{(\alpha,\beta)} \subset \mathcal{K}_{(\alpha,\beta)}$ generated by $H, B_a, P_a$. In other words, the rotations are redundant and we will ignore them for the purposes of showing that the coordinates $(t, x^a)$ are indeed global.

We now define a transitive action of $\mathcal{B}_{(\alpha,\beta)}$ on $\mathbb{R}^{d+1}$ by demanding that the map $j$ be equivariant. The action of $\mathcal{B}_{(\alpha,\beta)}$ on $\mathcal{M}_{(\alpha,\beta)}$ is induced by left multiplication on the group, so calculating the product of exponentials we arrive at:

$$
\begin{aligned}
e^{aH} \cdot (t, x^a) &= (t + a, x^a), \\
e^{v^a B_a} \cdot (t, x^a) &= \left( t, x^a + e^{-t\beta/2} \frac{\sin(\omega t)}{\omega} v^a \right), \\
e^{y^a P_a} \cdot (t, x^a) &= \left( t, x^a + e^{-t\beta/2} \left( \cos(\omega t) - \beta \frac{\sin(\omega t)}{2\omega} \right) y^a \right),
\end{aligned}
\tag{2.7}
$$

where $\omega := \frac{1}{2}\sqrt{4\alpha - \beta^2}$. This action is transitive almost by definition. Indeed, starting from the origin $(0, \mathbf{0})$ we can reach $(t, x^a)$ by acting with $\sigma(t, x^a) = e^{tH} e^{x^a P_a}$. By equivariance, the image of the map $j$ is an orbit of $\mathcal{B}_{(\alpha,\beta)}$ on $\mathcal{M}_{(\alpha,\beta)}$, but since $\mathcal{M}_{(\alpha,\beta)}$ is homogeneous, $j$ is surjective and, being a local diffeomorphism, it is a covering map.[4] Since both $\mathbb{R}^{d+1}$ and $\mathcal{M}_{(\alpha,\beta)}$ are simply connected, the map $j$ must be a diffeomorphism.

In summary, the coordinates $(t, x^a)$ thus defined are global and hence, as manifolds, all the spacetimes $\mathcal{M}_{(\alpha,\beta)}$ are diffeomorphic to $\mathbb{R}^{d+1}$. Either directly from equation (2.7) or starting with (2.5) via the method in [4], one calculates that the kinematical Lie algebra $\mathfrak{k}_{(\alpha,\beta)}$ is realized[5] on $\mathcal{M}_{(\alpha,\beta)}$ through the vector fields

$$
\begin{aligned}
\xi_{J_{ab}} &= x^b \partial_a - x^a \partial_b, \\
\xi_H &= \partial_t, \\
\xi_{P_a} &= e^{-\beta t/2} \left( \cos \omega t - \frac{\beta}{2} \frac{\sin \omega t}{\omega} \right) \partial_a, \\
\xi_{B_a} &= e^{-\beta t/2} \frac{\sin \omega t}{\omega} \partial_a.
\end{aligned}
\tag{2.8}
$$

---

[4]Strictly speaking it is a branched covering, but this cannot happen for equivariant maps between homogeneous spaces, since the existence of a non-empty branched locus would spoil homogeneity.

[5]Recall that in the standard conventions this is an *anti*-homomorphism of Lie algebras.

Note that depending on the respective values of $\alpha, \beta$ the parameter $\omega$ can be either real or imaginary, in all cases the vector fields above remain real and well defined.

Remark that although the time translations are simply a shift of the time coordinate $t$ by a constant, this is not the case for the spatial translations which are a shift with a particular function of time in case $\alpha$ or $\beta$ are non-zero.

Although as manifolds all the kinematical homogeneous spacetimes of galilean type $\mathcal{M}_{(\alpha,\beta)}$ are the same, this is no longer true if we equip them with invariant geometric structures, since the symmetries act differently in the different cases. We'll discuss this in more detail in the next subsection.

Before we introduce these additional geometric invariants, let us comment on the redundancy in the parameters $(\alpha, \beta)$. Observe that the change of basis $H \to sH, B_a \to s^{-1}B_a$, for any $s \in \mathbb{R}^{\times}$, leads to the change of parameters

$$\alpha \to s^2 \alpha, \qquad \beta \to s\beta. \tag{2.9}$$

This implies that the homogeneous spaces of which the parameters $(\alpha, \beta)$ are related through such a transformation are isomorphic.[6] Indeed this can be seen explicitly by observing that if one accompanies the transformation (2.9) with the coordinate transformation $(t, x^a) \to (s^{-1}t, x^a)$ then the vector fields (2.8) remain invariant.

Without loss of generality one can thus reduce the values of $(\alpha, \beta)$ via (2.9) to one of the following three classes

- Galilei spacetime: $\mathcal{M}_{(0,0)}$,

- (torsional) galilean dS spacetime: $\mathcal{M}_{(\gamma,1+\gamma)} \cong \mathcal{M}_{(\gamma s^2,(1+\gamma)s)}, \qquad \gamma \in [-1,1]$,

- (torsional) galilean AdS spacetime: $\mathcal{M}_{(1+\chi^2,2\chi)} \cong \mathcal{M}_{((1+\chi^2)s^2,2\chi s)}, \quad \chi \in [0,\infty)$.

The standard, rather well-known cases all have $\beta = 0$ and are Galilei spacetime ($\alpha = 0$), galilean de Sitter ($\alpha < 0$) and galilean Anti de Sitter ($\alpha > 0$). The cases with $\beta \neq 0$ are less well studied and are the topic of this paper. As we'll see in the next subsection, $\beta$ parameterizes the torsion of the unique invariant connection compatible with the invariant Newton–Cartan structure on the $\mathcal{M}_{(\alpha,\beta)}$.

In the following we will find it convenient to keep working with the parameters $(\alpha, \beta)$, rather than $\gamma$ or $\chi$ that parametrize the inequivalent spacetimes, since it will allow us to discuss all cases at once in a simple way. Furthermore each of $\alpha$ and $\beta$ will turn out to have a clear physical interpretation in the particle dynamics, with $\alpha$ parameterizing a harmonic potential and $\beta$ determining the damping.

## 2.3 The invariant NC-compatible connection

The difference between the various kinematical homogeneous spacetimes of galilean type $\mathcal{M}_{(\alpha,\beta)}$ is most explicitly seen in their unique invariant NC-compatible connection. This is the unique invariant affine connection that preserves the invariant Newton–Cartan structure on $\mathcal{M}_{(\alpha,\beta)}$, as we'll explain below. In modified exponential coordinates (2.5) the non-vanishing components of the invariant NC-compatible connection on $\mathcal{M}_{(\alpha,\beta)}$ are

$$\begin{aligned} \Gamma^a_{bt} &= \beta \delta^a_b, \\ \Gamma^a_{tt} &= \alpha x^a. \end{aligned} \tag{2.10}$$

Note that since $\Gamma^a_{tb} = 0$ this connection has torsion, the non-vanishing components being

$$T^a_{bt} = \beta \delta^a_b. \tag{2.11}$$

---

[6]As homogeneous spaces, not just as manifolds.

Furthermore the only non-vanishing components of the Riemann tensor are

$$R^a{}_{tbt} = -R^a{}_{ttb} = \alpha \delta^a_b. \tag{2.12}$$

We thus see that the parameters $(\alpha, \beta)$ can be identified with respectively the curvature and torsion of the unique invariant NC-compatible connection on $\mathcal{M}_{(\alpha,\beta)}$.

To derive the above results one starts from the fact [4] that since the $\mathcal{M}_{(\alpha,\beta)}$ are of galilean type, they carry an invariant Newton–Cartan structure[7] $(\tau_\mu, h^{\mu\nu})$. (In fact, one can rescale $\tau$ and $h$ independently, by nonzero real numbers, so one has a two-parameter family of invariant Newton–Cartan structures.) In the modified exponential coordinates $(x^\mu) = (t, x^a)$ introduced in (2.5), the invariant Newton–Cartan structure takes the same trivial form on all $\mathcal{M}_{(\alpha,\beta)}$:

$$\tau_\mu = \delta^t_\mu, \qquad h^{\mu\nu} = \delta^\mu_a \delta^\nu_b \delta^{ab}. \tag{2.13}$$

Crucially however the invariant affine connections differ significantly in the various cases. Such connections can be classified using the theorem of Nomizu [22], or more explicitly by demanding invariance under the infinitesimal coordinate transformations associated to the kinematical symmetries.[8] Both methods agree, and one finds that in modified exponential coordinates the only non-zero components of an invariant affine connection $\Gamma^\rho_{\mu\nu}$ on $\mathcal{M}_{(\alpha,\beta)}$ are[9]

$$
\begin{aligned}
\Gamma^t_{tt} &= (\kappa + \iota), \\
\Gamma^a_{tb} &= \kappa \delta^a_b, \\
\Gamma^a_{bt} &= (\beta + \iota)\delta^a_b, \\
\Gamma^a_{tt} &= \alpha x^a.
\end{aligned}
\tag{2.14}
$$

In summary, on a given $\mathcal{M}_{(\alpha,\beta)}$ there is a family of invariant connections parameterized by the two (unconstrained, real) constants $\kappa, \iota$.

A short calculation reveals that among this two parameter family of invariant connections there is a unique one that is compatible with the invariant Newton–Cartan structure (2.13) on $\mathcal{M}_{(\alpha,\beta)}$, i.e. such that $\nabla_\mu \tau_\nu = 0$ and $\nabla_\mu h^{\nu\rho} = 0$. This connection, that we refer to as the *invariant NC-compatible connection*, is the invariant affine connection for which $\kappa = \iota = 0$, which leads to (2.10).

## 3 Torsional galilean particles and the damped harmonic oscillator

In this section we will provide a physical interpretation to the less familiar galilean kinematical algebras with $\beta \neq 0$, by realizing them as the symmetries of free, or geodesic, particle motion in the space-time $\mathcal{M}_{(\alpha,\beta)}$. In the first part of this section we show how this geodesic motion with respect to the invariant NC compatible connection on $\mathcal{M}_{(\alpha,\beta)}$ can be identified, upon fixing time parametrization invariance, with the dynamics of the damped harmonic oscillator. In the second part of this section we study how (an extension of) the kinematical symmetry algebra $\mathfrak{k}_{(\alpha,\beta)}$ is realized in terms of phase space vector fields and conserved charges.

### 3.1 Damped harmonic oscillator from geodesic equation

To formulate a $\mathcal{K}_{(\alpha,\beta)}$ invariant particle dynamics on $\mathcal{M}_{(\alpha,\beta)}$ we can simply define the particle motion to be the geodesic equation with respect to an invariant connection. Since such invariant connections are not unique, see Section 2.3, there could a priori be more than one invariant

---

[7]See appendix A for a definition, nomenclature and some related notions.

[8]This amounts to solving $\mathcal{L}_\xi \nabla = 0$ for each $\xi$ in (2.8), which relative to local coordinates becomes the differential relations $\xi^\rho \partial_\rho \Gamma^\lambda_{\mu\nu} - \Gamma^\rho_{\mu\nu} \partial_\rho \xi^\lambda + \Gamma^\lambda_{\rho\nu} \partial_\mu \xi^\rho + \Gamma^\lambda_{\mu\rho} \partial_\nu \xi^\rho + \partial_\mu \partial_\nu \xi^\lambda = 0$.

[9]The result (2.14) is complete for $d \geq 3$. Some exceptions are present when $d \leq 2$, see Appendix C.

particle dynamics. It turns out however that the free parameters $\kappa, \iota$ specifying the different invariant connections drop out of the actual geodesic equations. In other words, all invariant connections share the same set of geodesics, something which should not really be a surprise, since in the highly symmetric situation we are considering one would expect[10] the geodesics to be generated purely by the symmetries, and these are independent of the parameters $\kappa$ and $\iota$.

The affine geodesic – or autoparallel – equation associated to an affine connection $\Gamma^{\rho}_{\mu\nu}$ reads

$$\frac{d^2 x^{\rho}}{d\sigma^2} + \Gamma^{\rho}_{\mu\nu} \frac{dx^{\mu}}{d\sigma} \frac{dx^{\nu}}{d\sigma} = f \frac{dx^{\rho}}{d\sigma}. \tag{3.1}$$

Choosing coordinates $(x^{\mu}) = (t, x^a)$ we can re-express $f$ via the $t$ component of the above equation:

$$f = \frac{\frac{d^2 t}{d\sigma^2} + \Gamma^t_{\mu\nu} \frac{dx^{\mu}}{d\sigma} \frac{dx^{\nu}}{d\sigma}}{\frac{dt}{d\sigma}}. \tag{3.2}$$

Inserting this into (3.1) and then specializing to the invariant connections (2.14) we find that it is equivalent to

$$\frac{d^2 x^a}{d\sigma^2} + \beta \frac{dx^a}{d\sigma} \frac{dt}{d\sigma} + \alpha x^a \left(\frac{dt}{d\sigma}\right)^2 - \frac{\frac{d^2 t}{d\sigma^2}}{\frac{dt}{d\sigma}} \frac{dx^a}{d\sigma} = 0. \tag{3.3}$$

As we mentioned at the beginning of this section the Nomizu parameters $\kappa$ and $\iota$ drop out of this equation, so that the geodesic equation is the same for all invariant connections. Upon choosing time $t$ to be the parameter along the curve,[11] i.e. $\sigma = t$, the equation (3.3) further simplifies to

$$\ddot{x}^a + \beta \dot{x}^a + \alpha x^a = 0, \tag{3.4}$$

with $\frac{d}{dt}$ denoted by a dot. Here, we recognize the dynamical equation of damped harmonic motion, with damping $\beta$ and stiffness $\alpha$.

We have thus arrived at a somewhat surprising but very concrete physical interpretation of particle motion on the most general galilean homogeneous kinematical spacetimes. Just as the invariant geodesics on Galilei spacetime correspond to free mechanical motion, the invariant geodesics on the spacetime $\mathbb{M}_{(\alpha,\beta)}$ correspond to damped harmonic motion. The invariant geodesics of the torsional galilean AdS spacetimes correspond to the underdamped cases ($\alpha > \frac{\beta^2}{4}$), while those of the torsional galilean dS spacetimes are equivalent to the overdamped ones[12] ($\alpha < \frac{\beta^2}{4}$). The critically damped oscillator describes the geodesics on the spacetime $\mathbb{M}_{(1,2)}$, which is the boundary case $\gamma = 1$ of the torsional galilean dS spacetimes.

## 3.2 The kinematical algebra through conserved charges

Since the equation (3.4) is equivalent to the geodesic equation of a $\mathcal{K}_{(\alpha,\beta)}$ invariant connection on $\mathbb{M}_{(\alpha,\beta)}$ it will exhibit $\mathcal{K}_{(\alpha,\beta)}$ symmetry by construction. To study this symmetry and the corresponding conserved charges it will be convenient to introduce an action that reproduces the damped harmonic dynamics (3.4) as its Euler-Lagrange equations. This action, known as the Bateman–Caldirola–Kanai (BCK) action [23–25], is

$$S = \int \frac{m e^{\beta t}}{2} (\dot{x}^a \dot{x}^a - \alpha x^a x^a) dt. \tag{3.5}$$

---

[10]Indeed this is the case.

[11]Note that if we choose $\sigma = t$ then $f = \Gamma^t_{tt} = \kappa + \iota$. So $t$ is not an affine parameter if one works with a connection for which $\kappa + \iota \neq 0$. Since this sum vanishes for the invariant NC-compatible connection (2.10), $t$ is an affine parameter in that case.

[12]Note that in these we include the $\alpha < 0$ range, where the potential is repulsive.

Here we introduced $m$, the mass of the particle, to provide a proper physical interpretation.

Since we chose time $t$ as our parameter,[13] it will be useful to describe transformations in their passive form. I.e. given a vector field $\xi^\mu$, so that $\delta x^\mu = \xi^\mu$, we define the passive transformations of the $x^a(t)$ as

$$\delta_{\text{passive}} x^a = \xi^a - \dot{x}^a \xi^t, \qquad \delta_{\text{passive}} \dot{x}^a = \frac{d}{dt} \delta_{\text{passive}} x^a, \qquad \delta_{\text{passive}} t = 0. \qquad (3.6)$$

The vector fields (2.8) that generated the $\mathcal{K}_{(\alpha,\beta)}$ action then lead to the following passive transformations

$$\delta_{J_{ab}} x^c = x^b \delta_a^c - x^a \delta_b^c, \qquad \delta_{P_a} x^b = \dot{F}(t) \delta_a^b, \qquad \delta_{B_a} x^b = F(t) \delta_a^b, \qquad \delta_H x^a = -\dot{x}^a, \quad (3.7)$$

with

$$F(t) = e^{-\beta t/2} \frac{\sin \omega t}{\omega}. \qquad (3.8)$$

For calculational purposes, it is useful to note that $F$ itself is a solution to the damped harmonic equation (3.4): $\ddot{F} + \beta \dot{F} + \alpha F = 0$. One checks that the commutators of the transformations (3.7) reproduce those of the vector fields (2.8) and thus also represent the Lie algebra $\mathfrak{k}_{(\alpha,\beta)}$.

The transformations corresponding to the rotations, boosts and spatial translations are symmetries in the most standard sense, in that they leave the action (3.5) invariant. Noether's theorem then leads to associated conserved charges. The time translation is a bit more subtle, as it rescales the action with a constant, i.e. $\delta_H S = \beta S$, rather than leaving it strictly invariant. This is sufficient however to leave the equations of motion invariant and so time translations are indeed, as was warranted by construction, also a symmetry of the particle motion. The association of a conserved charge is now a bit more subtle, but can still be performed.

The discussion of the conserved charges, as well as the Lie algebra they form, is clearest in the Hamiltonian formalism. It allows us to realize the kinematical algebra $\mathfrak{k}_{(\alpha,\beta)}$ as a Lie algebra of phase space vector fields that can then be made homomorphic to an algebra of conserved charges. The fact that the time translations rescale the action rather than leave it invariant (when $\beta \neq 0$), has two interesting related effects: the algebra of conserved charges is realized through a slight generalization of the standard Poisson bracket, and simultaneously the Bargmann extension – which appears for all Lie algebras $\mathfrak{k}_{(\alpha,\beta)}$ – is no longer central.

First one observes that the canonical momenta defined by the Lagrangian (3.5) are

$$p_a = m e^{\beta t} \dot{x}^a, \qquad (3.9)$$

and one finds the canonical Hamiltonian

$$h = \frac{e^{-\beta t}}{2m} p_a p_a + \frac{m \alpha e^{\beta t}}{2} x^a x^a. \qquad (3.10)$$

Via (3.9) one finds the (on-shell) transformations of the canonical momenta and by introducing the phase space coordinates $(y^A) = (x^a, p_a)$ one can then express the transformations on phase space in terms of phase space vector fields, $\Xi = \Xi^a \partial_a + \Xi_a \partial^a$, via $\delta y^A = \Xi^A$. Here we used the notation $\partial^a = \frac{\partial}{\partial p_a}$. Carrying out this procedure for all symmetry transformations one finds

$$\Xi_{J_{ab}} = x^b \partial_a - x^a \partial_b + p_b \partial^a - p_a \partial^b, \qquad (3.11)$$

$$\Xi_{P_a} = \dot{F} \partial_a + m e^{\beta t} \ddot{F} \partial^a, \qquad (3.12)$$

$$\Xi_{B_a} = F \partial_a + m e^{\beta t} \dot{F} \partial^a, \qquad (3.13)$$

$$\Xi_H = -\frac{e^{-\beta t}}{m} p_a \partial_a + (m \alpha e^{\beta t} x^a + \beta p_a) \partial^a, \qquad (3.14)$$

---

[13]A fully reparametrization and coordinate invariant formulation is discussed in section 4. Although we do not discuss the Hamiltonian formulation there, a discussion identical to the one in this section can be repeated keeping reparametrization invariance manifest, by using the formalism of constrained Hamiltonian systems.

where $F$ is given by equation (3.8). One verifies that under commutation the above phase space vector fields close into an algebra which is (anti-)isomorphic to the kinematical algebra $\mathfrak{k}_{(\alpha,\beta)}$ defined in (2.1) and (2.2).

We should now recall the notion of hamiltonian vector field. The canonical symplectic form is $\Omega = dx^a \wedge dp_a$. A vector field $X$ is hamiltonian if there exists a phase space function $f$ such that

$$i_X \Omega = df \qquad \Leftrightarrow \qquad X^a = \partial^a f, \quad X_a = -\partial_a f. \tag{3.15}$$

In this case, we will write $X = X_f$. It follows that the commutator of two hamiltonian vector fields can be re-expressed in terms of the Poisson bracket:

$$[X_{f_1}, X_{f_2}] = -X_{\{f_1, f_2\}}. \tag{3.16}$$

In particular, the time evolution is determined by the hamiltonian vector field associated to the canonical Hamiltonian (3.10):

$$X_h = \frac{e^{-\beta t}}{m} p_a \partial_a - m\alpha e^{\beta t} x^a \partial^a, \tag{3.17}$$

via Hamilton's equations

$$\frac{dy^A}{dt} = X_h^A. \tag{3.18}$$

These imply the following time evolution for an arbitrary (time-dependent) phase space function $f$:

$$\frac{d}{dt} f = \partial_t f + \{f, h\}. \tag{3.19}$$

It follows that a phase space function $f$ is conserved when $\{h, f\} = \partial_t f$, or

$$[X_f, X_h] = X_{\partial_t f}. \tag{3.20}$$

The vector fields (3.11-3.13) associated to rotations, spatial translations and boosts are all hamiltonian:

$$\Xi_{J_{ab}} \;=\; X_{J_{ab}}, \qquad \text{with} \quad J_{ab} = x^b p_a - x^a p_b, \tag{3.21}$$

$$\Xi_{P_a} \;=\; X_{P_a}, \qquad \text{with} \quad P_a = \dot{F} p_a - m e^{\beta t} \ddot{F} x^a, \tag{3.22}$$

$$\Xi_{B_a} \;=\; X_{B_a}, \qquad \text{with} \quad B_a = F p_a - m e^{\beta t} \dot{F} x^a, \tag{3.23}$$

where $F$ is given by equation (3.8). One checks that all of $J_{ab}$, $P_a$ and $B_a$ satisfy (3.20) and are hence conserved charges. They form the following Poisson algebra

$$\{J_{ab}, J_{cd}\} \;=\; \delta_{bc} J_{ad} - \delta_{ac} J_{bd} - \delta_{bd} J_{ac} + \delta_{ad} J_{bc}, \tag{3.24}$$

$$\{J_{ab}, B_c\} \;=\; \delta_{bc} B_a - \delta_{ac} B_b, \qquad \{J_{ab}, P_c\} = \delta_{bc} P_a - \delta_{ac} P_b, \tag{3.25}$$

$$\{P_a, B_b\} \;=\; m \delta_{ab}. \tag{3.26}$$

The first two lines, (3.24, 3.25), reproduce the brackets of the subalgebra of the kinematical algebra $\mathfrak{k}_{(\alpha,\beta)}$ spanned by rotations, boosts and spatial translations, while the last line provides the well known Bargmann extension [9]. The mass $m$ is a constant, which implies that $X_m = 0$ and that $m$ Poisson commutes with all the $J_{ab}$, $P_a$ and $B_a$ – so that the extension (3.26) is central in the subalgebra (3.24, 3.25). It is interesting to note that although the charges intricately depend on the parameters $(\alpha, \beta)$ and time $t$, see (3.21-3.23, 3.8), the Bargmann extension is universal to all cases and independent of these parameters. In deriving (3.26) the relation $\dot{F}^2 - \ddot{F}F = e^{-\beta t}$ is crucial, it expresses the Wronskian of $F$ and $\dot{F}$, both of which are solutions to the dynamical equation (3.4).

We have reserved the discussion of the time translation symmetry and the associated vector field $\Xi_H$ until now, as it is more subtle. Indeed, the vector field $\Xi_H$ (3.14) is not hamiltonian, indeed not even symplectic. Instead, it rescales the symplectic structure by a constant:

$$\mathcal{L}_{\Xi_H}\Omega = \beta\,\Omega\,. \tag{3.27}$$

We discuss such symplectic homotheties and how they can lead to conserved charges in more generality and detail in Appendix B. Here we simply remark that the difference between $\Xi_H$ and $\beta E$ is however hamiltonian, where we introduced a preferred symplectic homothety

$$E = \frac{1}{2}(x^a\partial_a + p_a\partial^a)\,. \tag{3.28}$$

In other words, we can write

$$\Xi_H = X_{(\beta,H)} := \beta E + X_H\,, \tag{3.29}$$

where a short computation reveals that

$$H = -\left(\frac{e^{-\beta t}}{2m}p_a p_a + \frac{\beta}{2}x^a p_a + \frac{m\alpha e^{\beta t}}{2}x^a x^a\right) = -\left(h + \frac{\beta}{2}x^a p_a\right)\,. \tag{3.30}$$

Interestingly $H$ is again a conserved charge. This can simply be verified by direct computation of $\frac{d}{dt}H$ and use of the equations of motion. This is however not a coincidence: in Appendix B we show how for any homothetic hamiltonian vector field $X_{(s,f)} = sE + X_f$ that satisfies

$$\mathcal{L}_{X_{(s,f)}}h = sh + \partial_t f\,, \qquad \mathcal{L}_E h = h\,, \qquad \mathcal{L}_E\Omega = \Omega\,, \tag{3.31}$$

the phase space function $f$ is a conserved charge. This is a generalization of the standard argument for hamiltonian vector fields $X_f = X_{(0,f)}$ to homothetic hamiltonian vector fields $X_{(s,f)}$. Verifying that indeed (3.28) and (3.29) satisfy the conditions (3.31), with $s = \beta$ and $f = H$, is thus another way of showing that $H$ is a conserved charge.

It is now important to point out that although the homothetic hamiltonian vector fields still form a subalgebra of the algebra of vector fields (see Appendix B), this algebra is no longer homomorphic to the Poisson algebra of phase space functions. Rather one finds the relation

$$[X_{(s_1,f_1)},X_{(s_2,f_2)}] = -[\![(s_1,f_1),(s_2,f_2)]\!]\,, \tag{3.32}$$

where

$$[\![(s_1,f_1),(s_2,f_2)]\!] := (0,\{f_1,f_2\} - s_1(\mathcal{L}_E f_2 - f_2) + s_2(\mathcal{L}_E f_1 - f_1))\,. \tag{3.33}$$

It should be clear that (3.32) is a direct generalization of (3.16) and that furthermore $[\![(0,f_1),(0,f_2)]\!] = (0,\{f_1,f_2\})$. Note however that although $[\![\cdot,\cdot]\!]$ is a Lie bracket, it is *not* a Poisson bracket (nor even a Jacobi bracket [26,27]). An important subtlety that will play a role below, is that the constant phase space functions $c$, are no longer central (as they are in the Poisson algebra):

$$[\![(s,f),(0,c)]\!] = (0,sc)\,. \tag{3.34}$$

Since it is the algebra of phase space vector fields (3.11-3.14) that is (anti-) isomorphic to the kinematical algebra $\mathfrak{k}_{(\alpha,\beta)}$ and since one of these vector fields is homothetic hamiltonian rather than simply hamiltonian, it follows that we will be able to recover the kinematical algebra in terms of conserved charges only with respect to the generalized bracket (3.33) and

by taking into account the scaling weight $s$, defined via $\mathcal{L}_\Xi \Omega = s\Omega$, of each symmetry. Indeed, an explicit calculation shows that the non-vanishing brackets are

$$
\begin{aligned}
[\![\mathbf{J}_{ab}, \mathbf{J}_{cd}]\!] &= \delta_{bc}\mathbf{J}_{ad} - \delta_{ac}\mathbf{J}_{bd} - \delta_{bd}\mathbf{J}_{ac} + \delta_{ad}\mathbf{J}_{bc}\,, \\
[\![\mathbf{J}_{ab}, \mathbf{B}_c]\!] &= \delta_{bc}\mathbf{B}_a - \delta_{ac}\mathbf{B}_b\,, \\
[\![\mathbf{J}_{ab}, \mathbf{P}_c]\!] &= \delta_{bc}\mathbf{P}_a - \delta_{ac}\mathbf{P}_b\,, \\
[\![\mathbf{H}, \mathbf{B}_a]\!] &= -\mathbf{P}_a\,, \\
[\![\mathbf{H}, \mathbf{P}_a]\!] &= \alpha\mathbf{B}_a + \beta\mathbf{P}_a\,, \\
[\![\mathbf{P}_a, \mathbf{B}_b]\!] &= \delta_{ab}\mathbf{M}\,, \\
[\![\mathbf{H}, \mathbf{M}]\!] &= \beta\,\mathbf{M}\,,
\end{aligned}
\tag{3.35}
$$

where we introduced the shorthands

$$
\mathbf{J}_{ab} = (0, J_{ab}), \quad \mathbf{P}_a = (0, P_a), \quad \mathbf{B}_a = (0, B_a), \quad \mathbf{H} = (\beta, H) \quad \text{and} \quad \mathbf{M} = (0, m). \tag{3.36}
$$

The Lie algebra (3.35) is a one-dimensional extension, by the generator $\mathbf{M}$, of the galilean kinematical algebra $\mathfrak{k}_{(\alpha,\beta)}$ defined in (2.1) and (2.2). In case $\beta = 0$ one recovers the Bargmann central extensions of the Galilei ($\alpha = 0$) and Newton–Hooke ($\alpha > 0$ and $\alpha < 0$) algebras. If $\beta \neq 0$, the appearance of the mass $\mathbf{M}$ on the right hand side of the bracket of boosts and commutators remains intact, but this Bargmann extension is now no longer central, as can be seen from the nonzero $[\![\mathbf{H}, \mathbf{M}]\!]$ bracket. Physically this has the interpretation of time translations rescaling the mass of the particle by a constant. Indeed, the physical mass of the particle can be identified with the overall prefactor of the action. Since time-translations rescale the action rather than leaving it invariant when $\beta \neq 0$, the effect of a time translation is indeed to rescale the physical mass.

For any values of $\alpha, \beta$ the Lie algebras (3.35) are a deformation of the centrally extended static kinematical Lie algebra. Such deformations were classified in [16, Table 2] for $d = 3$ and [17, Table 18] for $d > 3$. In section 4.3 we show how they also appear as the algebras of homotheties of the Lorentzian metrics (4.23), that are obtained through the Eisenhart lift of the damped harmonic oscillator. They are also the isometry Lie algebras of certain homogeneous pp-waves discussed in [15], as we explain in appendix D.

We end this section with a small curiosity. As we discussed above, the conserved charges form a Lie algebra homomorphic to the kinematical algebra only upon the introduction of a modified bracket. Somewhat surprisingly the charges also close under the usual Poisson bracket. In this case they do not reproduce the kinematical algebra $\mathfrak{k}_{(\alpha,\beta)}$, but rather a central extension of the kinematical algebra at *different* values of the parameters, i.e. $\mathfrak{k}_{(\omega^2,0)}$.

To see this explicitly one can first compute

$$
\{H, B_a\} = -P_a - \frac{\beta}{2}B_a\,, \qquad \{H, P_a\} = \alpha B_a + \frac{\beta}{2}P_a\,. \tag{3.37}
$$

Upon performing the change of basis

$$
\tilde{P}_a = P_a + \frac{\beta}{2}B_a\,, \tag{3.38}
$$

one then gets the following Poisson bracket algebra:

$$
\begin{aligned}
\{J_{ab}, J_{cd}\} &= \delta_{bc}J_{ad} - \delta_{ac}J_{bd} - \delta_{bd}J_{ac} + \delta_{ad}J_{bc}\,, \\
\{J_{ab}, B_c\} &= \delta_{bc}b_a - \delta_{ac}b_b\,, \\
\{J_{ab}, \tilde{P}_c\} &= \delta_{bc}\tilde{P}_a - \delta_{ac}\tilde{P}_b\,, \\
\{H, B_a\} &= -\tilde{P}_a\,, \\
\{H, \tilde{P}_a\} &= \omega^2 B_a\,, \\
\{\tilde{P}_a, B_b\} &= m\,\delta_{ab}\,,
\end{aligned}
\tag{3.39}
$$

which is the Bargmann central extension of the galilean kinematical algebra $\mathfrak{k}_{(\omega^2,0)}$. This shows that the free motion of a particle on $\mathcal{M}_{(\alpha,\beta)}$, or equivalently a damped harmonic oscillator of parameters $\alpha,\beta$, has not only $\mathfrak{k}_{(\alpha,\beta)}$ symmetry, but also $\mathfrak{k}_{(\omega^2,0)}$ symmetry. It is well known, see e.g. [12], that the algebra of all symmetries of the damped harmonic oscillator is larger than $\mathfrak{k}_{(\alpha,\beta)}$, but what is somewhat surprising is that this larger symmetry group apparently also has $\mathfrak{k}_{(\omega^2,0)}$ as a subgroup.

# 4  Particle motion via Newton-Cartan and Bargmann geometry

In the previous section we saw how a free particle, or from a mathematical point of view a geodesic, on a galilean kinematical spacetime $\mathcal{M}_{(\alpha,\beta)}$ is equivalent to damped harmonic motion. This is made explicit by working in adapted coordinates $(t, x^a)$ that split time and space and furthermore by choosing time $t$ as the parameter along the geodesic. In this section we return to a more intrinsic and geometric description by rewriting the action (3.5) in a coordinate and reparametrization invariant way. This will make use of a Newton-Cartan structure, which – somewhat surprisingly – is not invariant, but only homothetically invariant under $\mathcal{K}_{(\alpha,\beta)}$. Indeed, since the action for a particle on a Newton-Cartan background leads to a compatible symmetric connection, but the analysis of Section 2.3 revealed that when $\beta \neq 0$ the invariant compatible connection is not symmetric, an action based on the invariant Newton-Cartan structure on $\mathcal{M}_{(\alpha,\beta)}$ cannot lead to a geodesic equation involving an invariant connection (when $\beta \neq 0$).

We will first, in section 4.1 review the general formulation and properties of the action describing particle dynamics on a Newton-Cartan background and then, in subsection 4.2, specialize to the case of our interest: $\mathcal{K}_{(\alpha,\beta)}$ invariant motion on $\mathcal{M}_{(\alpha,\beta)}$. Finally, we will discuss in subsection 4.3 how the Bargmann extended algebra, that appeared through a careful consideration of the conserved charges on phase space in section 3.2, can be given a geometric interpretation as an algebra of homotheties[14] of an associated (higher dimensional) Lorentzian metric through the Eisenhart lift.

## 4.1  Particle motion on Newton–Cartan spacetimes

A Newton–Cartan spacetime is a $d + 1$ dimensional manifold $\mathcal{M}$ equipped with a Newton–Cartan structure $(\tau_\mu, h^{\mu\nu})$, see appendix A. Particle motion on such a Newton–Cartan spacetime is then described in terms of a curve $x^\mu(\sigma)$, where $x^\mu$ are coordinates on $\mathcal{M}$ and $\sigma$ is a worldline parameter. Free particle motion is essentially geodesic and so to define it one needs an affine connection. Contrary to the riemannian or lorentzian case there is more than one torsion free compatible connection for a given Newton–Cartan structure. This implies that to define particle motion one needs to provide additional information. A priory this could be (part of) the connection itself, but this turns out not to be the most useful way to package this additional freedom, especially in case one would want to formulate a variational principle. The action for a particle on a Newton–Cartan background goes back to [28], reviews from a more modern perspective can be found in e.g. [29, 30].

To write a reparametrization invariant particle Lagrangian, one introduces what we will refer to as a *compatible NC-doublet*, $(\hat{\tau}^\mu, \bar{h}_{\mu\nu})$, see Appendix A for a precise definition and further details.

---

[14]Actually the same transformations are also isometries of a conformally related metric, see appendix D.

A Newton–Cartan structure together with a compatible NC-doublet then define the action

$$S[x^\mu(\sigma)] = \frac{m}{2} \int \frac{\bar{h}_{\mu\nu}\frac{dx^\mu}{d\sigma}\frac{dx^\nu}{d\sigma}}{\tau_\rho\frac{dx^\rho}{d\sigma}}\, d\sigma\,. \tag{4.1}$$

In this paper we will furthermore restrict attention to the case $\partial_{[\mu}\tau_{\nu]} = 0$, the Euler–Lagrange equations for the action (4.1) then take the form

$$\frac{d^2 x^\mu}{d\sigma^2} + \Gamma^\mu_{\rho\sigma}\frac{dx^\rho}{d\sigma}\frac{dx^\sigma}{d\sigma} = \frac{\frac{d}{d\sigma}N}{N}\frac{dx^\mu}{d\sigma}\,, \tag{4.2}$$

where $N = \tau_\rho\frac{d}{d\sigma}x^\rho$ and

$$\Gamma^\lambda_{\mu\nu} = \hat{\tau}^\lambda\partial_{(\mu}\tau_{\nu)} + \frac{1}{2}h^{\lambda\rho}(\partial_\mu\bar{h}_{\nu\rho} + \partial_\nu\bar{h}_{\mu\rho} - \partial_\rho\bar{h}_{\mu\nu})\,. \tag{4.3}$$

As the notation suggests, the $\Gamma^\lambda_{\mu\nu}$ provide an affine connection on $\mathcal{M}$, which is symmetric and leaves the Newton–Cartan structure $(\tau_\mu, h^{\mu\nu})$ invariant.

We can thus conclude that particle motion on a Newton–Cartan spacetime $(\mathcal{M}, \tau_\mu, h^{\mu\nu})$, as specified by the action (4.1), is equivalent to geodesic motion with respect to a symmetric connection compatible with the Newton–Cartan structure. Both are determined by a choice of compatible NC-doublet $(\hat{\tau}^\mu, \bar{h}_{\mu\nu})$.

Let us now connect the somewhat abstract discussion above to the more familiar mechanics of a nonrelativistic particle. Under the assumption $\partial_{[\mu}\tau_{\nu]} = 0$, we can always (locally) make a choice of coordinates $(t, x^a)$ such that $\tau_\mu = \delta^t_\mu$. Additionally, we can choose the worldline parameter to coincide with our choice of time: $\sigma = t$. It follows that $h^{\mu\nu} = \delta^\mu_a\delta^\nu_b h^{ab}$ and we can choose (without loss of generality) a compatible NC-doublet $\hat{\tau}^\mu = \delta^\mu_t$, $\bar{h}_{\mu\nu} = \delta^a_\mu\delta^b_\nu h_{ab} + \delta^t_\mu C_\nu + \delta^t_\nu C_\mu$. The action (4.1) then takes the form

$$S[x^a(t)] = m \int \left(\frac{1}{2}h_{ab}\dot{x}^a\dot{x}^b + C_a\dot{x}^a + C_t\right)dt\,. \tag{4.4}$$

We recognize here the Lagrangian of a particle moving in a $d$-dimensional space with Riemannian metric $h_{ab}$, under the influence of a vector potential $C_a$ and scalar potential $-C_t$.

The motion of particles on the galilean homogeneous spacetimes $\mathcal{M}_{(\alpha,\beta)}$ are highly symmetric special cases of the above. In general these symmetries, which in this setting are symmetries of the action (4.1), are generated by infinitesimal diffeomorphisms on $\mathcal{M}$ that take the form

$$\delta_\xi x^\mu = \xi^\mu\,. \tag{4.5}$$

The Lagrangian (4.1) transforms under such a diffeomorphism as

$$\delta_\xi L = \frac{m}{2}\left(\frac{(\mathcal{L}_\xi\bar{h}_{\mu\nu})\frac{dx^\mu}{d\sigma}\frac{dx^\nu}{d\sigma}}{\tau_\mu\frac{dx^\mu}{d\sigma}} - \frac{\bar{h}_{\mu\nu}\frac{dx^\mu}{d\sigma}\frac{dx^\nu}{d\sigma}(\mathcal{L}_\xi\tau_\rho)\frac{dx^\rho}{d\sigma}}{(\tau_\mu\frac{dx^\mu}{d\sigma})^2}\right). \tag{4.6}$$

This implies that if $\xi^\mu$ satisfies

$$\mathcal{L}_\xi\tau_\mu = \zeta\tau_\mu\,, \tag{4.7}$$
$$\mathcal{L}_\xi\bar{h}_{\mu\nu} = (\lambda + \zeta)\bar{h}_{\mu\nu} + 2\tau_{(\nu}\partial_{\mu)}K\,,$$

for some arbitrary functions $K, \zeta$ and constant $\lambda$, then

$$\delta_\xi L = \lambda L + m\frac{d}{d\sigma}K\,. \tag{4.8}$$

The above is equivalent to the action (4.1) being invariant up to a rescaling with the constant $\lambda$. It follows that transformations of the form (4.5) with $\xi$ satisfying (4.7) leave the equations of motion (4.2) invariant so can be considered symmetries. The associated conserved Noether charges are

$$Q_\xi(\sigma) = \frac{\delta L}{\delta \frac{dx^\mu}{d\sigma}} \xi^\mu - mK - \lambda \int^\sigma L d\sigma'. \tag{4.9}$$

The generalization to symmetries that rescale the action rather than leave it invariant are crucial, since, as we discussed in Section 3.2, the time translations are of this type when $\beta \neq 0$.

## 4.2 Covariant description of motion on $\mathcal{M}_{(\alpha,\beta)}$

We can now view the action (3.5) as a special case of (4.4) and that way rewrite it as a manifestly covariant and reparametrization invariant particle action (4.1) on a Newton-Cartan spacetime. One verifies that the dynamic equation (3.3) describing the particle on $\mathcal{M}_{(\alpha,\beta)}$ equals the Euler–Lagrange equations (4.2) for the action (4.1) with Newton–Cartan structure

$$\tau_\mu = \delta^t_\mu, \qquad h^{\mu\nu} = e^{-\beta t} \delta^\mu_a \delta^\nu_a \tag{4.10}$$

and compatible NC-doublet

$$\hat{\tau}^\mu = \delta^\mu_t, \qquad \bar{h}_{\mu\nu} = e^{\beta t} \delta^a_\mu \delta^a_\nu - \alpha e^{\beta t} x^a x^a \delta^t_\mu \delta^t_\nu. \tag{4.11}$$

Observe that the Newton–Cartan structure (4.10) is *not* the invariant one (2.13) when $\beta \neq 0$. The above shows however that the *symmetric part*[15] of the unique invariant connection (2.10) compatible with the invariant Newton–Cartan structure (2.13) coincides with the symmetric connection (4.3) compatible with the Newton–Cartan structure (4.10) and specified by the NC-doublet (4.11).

The Newton–Cartan structure (4.10) is invariant under the subalgebra generated by the $\xi_{P_a}, \xi_{B_a}$ and $\xi_{J_{ab}}$ of (2.8), but this is not the case for the time translations: although $\mathcal{L}_{\xi_H} \tau_\mu = 0$ one finds that $\mathcal{L}_{\xi_H} h^{\mu\nu} = -\beta h^{\mu\nu}$. Similarly one verifies that the $\xi_{P_a}, \xi_{B_a}$ and $\xi_{J_{ab}}$ satisfy (4.7) for (4.10, 4.11) with $\lambda = \zeta = 0$ and

$$K_{J_{ab}} = 0, \qquad K_{P_a} = e^{\beta t} \ddot{F} x^a, \qquad K_{B_a} = e^{\beta t} \dot{F} x^a, \tag{4.12}$$

where $F$ was defined in (3.8). It follows that the transformations associated to the rotations, boosts and spatial translations leave the action invariant and are thus symmetries in the strictest sense. The time translation vector field $\xi_H$ also satisfies (4.7) for (4.10, 4.11), but now with

$$\lambda_H = \beta \quad \text{and} \quad K_H = \zeta_H = 0. \tag{4.13}$$

This implies the time-translations do not leave the action invariant, but rather rescale it with the constant $\beta$, they are therefore symmetries in a slightly weaker sense: they still leave the geodesic equation invariant – as indeed they should since that equation was constructed using the invariant connection. I.e. we come, as expected, to the same conclusion as in Section 3.

## 4.3 Eisenhart lift: the extended kinematical algebra through homotheties

About a century ago Eisenhart [11] pointed out that a large class of nonrelativistic mechanical systems can be equivalently described in terms of null geodesic motion on a higher dimensional lorentzian spacetime with a null Killing vector. This relation was rediscovered by Duval et

---

[15]Remark that it is only the symmetric part of a connection that appears in the geodesic equation (3.1).

al. [10] (see also [31–33]) and more recently studied from a more general perspective in [34, 35], among others. Looked at from a top down perspective this equivalence becomes a null reduction [10, 36].

From a geometric perspective, and restricted to the subclass of cases that we are interested in, the relation can be precisely phrased as the equivalence of $d + 2$ dimensional Bargmann structures with parallel null Killing vector and $d + 1$ dimensional torsionless Type I Newton–Cartan geometry.

Let us for clarity recall the relevant definitions. A Bargmann structure is a lorentzian metric $g_{AB}$ together with a nowhere vanishing null vector field $k^A$. This vector field is parallel when $\nabla_A k^B = 0$ (w.r.t. the Levi-Civita connection of $g_{AB}$) and is then also automatically Killing. Type I Newton–Cartan geometry can take various equivalent shapes. We'll define it to be a Newton–Cartan structure $(\tau_\mu, h^{\rho\sigma})$ together with an equivalence class of compatible NC doublets $[(\hat{\tau}^\mu, \bar{h}_{\rho\sigma})]$, see Appendix A for more details, it is torsionless when the intrinsic torsion [37] of the galilean $\mathcal{G}$-structure vanishes, i.e. $d\tau = 0$.

The equivalence between these two geometric concepts is most explicit upon a choice of coordinates $(x^A) = (u, x^\mu)$ such that $k = \partial_u$. The $(d + 2)$ dimensional lorentzian metric then takes the form

$$ds^2 = -2du\,\tau_\mu dx^\mu + \bar{h}_{\mu\nu} dx^\mu dx^\nu. \tag{4.14}$$

In components this implies

$$(g_{AB}) = \begin{pmatrix} 0 & -\tau_\mu \\ -\tau_\nu & \bar{h}_{\mu\nu} \end{pmatrix}, \qquad (g^{AB}) = \begin{pmatrix} -\hat{\tau}^\rho \hat{\tau}^\sigma \bar{h}_{\rho\sigma} & -\hat{\tau}^\mu \\ -\hat{\tau}^\nu & h^{\mu\nu} \end{pmatrix}. \tag{4.15}$$

A different choice of coordinates $u' = u + \Lambda(x^\mu)$ is equivalent to a change

$$\hat{\tau}'^\mu = \hat{\tau}^\mu - h^{\mu\nu}\partial_\mu \Lambda, \qquad \bar{h}'_{\mu\nu} = \bar{h}_{\mu\nu} + \partial_\mu \Lambda \tau_\nu + \partial_\nu \Lambda \tau_\mu, \tag{4.16}$$

which leaves the equivalence class invariant, i.e. $[(\hat{\tau}^\mu, \bar{h}_{\mu\nu})] = [(\hat{\tau}'^\mu, \bar{h}'_{\mu\nu})]$. Note that the identification (4.14) implies that $(k_A) = (0, \tau_\mu)$ and so $k$ being parallel becomes equivalent to $\tau$ being closed.

Eisenhart's equivalence between nonrelativistic motion and null geodesic motion is then a direct consequence of the geometric equivalence reviewed above. First, one observes that a null curve $(x^A(s)) = (u(s), x^\mu(s))$ for the metric (4.14) satisfies

$$\dot{u} = \frac{1}{2} \frac{\bar{h}_{\mu\nu} \dot{x}^\mu \dot{x}^\nu}{\tau_\rho \dot{x}^\rho}. \tag{4.17}$$

Using this relation (and that $\tau$ is closed), a short calculation then reveals that for such a null curve

$$\ddot{x}^A + \Gamma^A_{BC} \dot{x}^A \dot{x}^C = f\,\dot{x}^A \quad \Longleftrightarrow \quad \ddot{x}^\mu + \Gamma^\mu_{\rho\sigma} \dot{x}^\rho \dot{x}^\sigma = \frac{\dot{N}}{N} \dot{x}^\mu. \tag{4.18}$$

On the left side of the equivalence the connection is the Levi-Civita connection of the metric $g_{AB}$, while on the right hand side the connection is the Newton–Cartan one (4.3), and one recognizes the equation (4.2) which, as discussed in section 4.1, via the choices $(x^\mu) = (t, x^a)$, $s = t$ describes the motion of a nonrelativistic particle described by the action (4.4). Interestingly, the relation (4.17) also provides a geometric interpretation of the NC action (4.1), i.e. $S_{NC} = \Delta u$, or in other words: Nonrelativistic particle motion follows null geodesics upon the Eisenhart lift and furthermore such null geodesics extremize the coordinate difference $\Delta u$ between start and endpoint. Note that this is due to the Bargmann structure exhibited by the lifted geometry and remark that $\Delta u = \Delta \tilde{u}$ is indeed a geometric invariant (although we expressed it in local coordinates).

Possibly the most interesting feature of the Eisenhart lift is that it provides a new, more geometric, perspective on the symmetries of the mechanical system [10]. To start the discussion we recall that the right-hand side of (4.17) is the nonrelativistic Lagrangian, i.e. $m\dot{u} = L$. Those symmetries of the lower dimensional nonrelativistic system that leave the Lagrangian invariant only up to the addition of a total derivative, i.e. $\delta L = m\dot{K}$, will thus have to act on $u$ as well. In particular, to leave the relation (4.17) invariant one has to impose $\delta u = K$. If we let $\xi^\mu$ be the $(d+1)$ dimensional vector field generating the nonrelativistic symmetry, i.e. $\delta x^\mu = \xi^\mu$, then its lift to $(d+2)$ dimensions will thus be $(\hat{\xi}^A) = (K, \xi^\mu)$. It follows that the $(d+2)$ dimensional bracket is $([\hat{\xi}_1, \hat{\xi}_2]^A_{d+2}) = (\xi_1 K_2 - \xi_2 K_1, [\xi_1, \xi_2]^\mu_{d+1})$. The $(d+1)$ dimensional vector fields form by assumption an algebra, i.e. $[\xi_1, \xi_2]_{d+1} = \xi_3$, and it thus follows that[16]

$$[\hat{\xi}_1, \hat{\xi}_2]_{d+2} = \hat{\xi}_3 + a k, \qquad a = (\xi_1 K_2 - \xi_2 K_1 - K_3), \ \xi_3 = [\xi_1, \xi_2]_{d+1}. \tag{4.19}$$

So if there are symmetries of the original nonrelativistic mechanical system that change the Lagrangian by a total derivative, then through the Eisenhart lift they extend the symmetry algebra by an extra generator $ak$.[17]

A striking example is already provided by the Galilei symmetry algebra $\mathfrak{k}_{(0,0)}$ of a nonrelativistic free particle. Taking $\xi_1 = \xi_P = \partial_x$ to be a translation and $\xi_2 = \xi_B = t\partial_x$ a boost, one finds that $\xi_3 = 0$, as well as $K_1 = K_3 = 0$ but $K_2 = x$. It follows that $[\hat{\xi}_P, \hat{\xi}_B]_{d+2} = k$. Furthermore $k$ commutes with all other generators $\hat{\xi}$ of the (lifted) Galilei symmetries and one thus recovers the Bargmann extension of the Galilei group. The Eisenhart lift thus provides a geometric realization of this classic central extension.

We are now ready to return to the main topic of this paper, namely the galilean kinematical algebras of generic type. In that case, as we discussed in Section 3.2 and Section 4.2, we need to take into account that some symmetries (in particular time translation) are of an even more general type, in that they also rescale the Lagrangian rather than only leaving it invariant up to a total derivative, i.e. $\delta L = \lambda L + m\dot{K}$. Via (4.17), this then implies $\delta u = \lambda u + K$ which leads to a more general form of the lift, i.e.,

$$\hat{\xi} = (\lambda u + K)\partial_u + \xi. \tag{4.20}$$

As a check, one verifies that if $\xi$ is a symmetry of the nonrelativistic dynamics, i.e. satisfies (3.31), then $\hat{\xi}$ is a conformal Killing vector of the Bargmann structure and hence a symmetry of the null-geodesic equation, i.e.

$$\mathcal{L}_{\hat{\xi}} g_{AB} = (\lambda + \zeta) g_{AB}. \tag{4.21}$$

Let us recall that, as mentioned in the previous subsection, $\zeta$ is zero for all vector fields (2.8) generating $\mathfrak{k}_{(\alpha,\beta)}$. In this case the lifted vector fields $\hat{\xi}$ then act as homotheties, since $\lambda$ is constant.

The algebra of the lifted vector fields is then

$$[\hat{\xi}_1, \hat{\xi}_2]_{d+2} = \hat{\xi}_3 + a k, \qquad a = (K_1 \lambda_2 - K_2 \lambda_1 + \mathcal{L}_{\xi_1} K_2 - \mathcal{L}_{\xi_2} K_1 - K_3), \ \xi_3 = [\xi_1, \xi_2]_{d+1}. \tag{4.22}$$

So apart from the small addition to the definition of $a$ with respect to (4.19), it might appear not much has changed. But that is not the case. Let us restrict attention to the case where $a$

---

[16]Note that, by definition, $\hat{\xi}_3 = K_3 \partial_u + \xi_3$. But $a$ is defined in a way that it contains $-K_3$ and thus $\hat{\xi}_3 + ak$ is independent of $K_3$.

[17]Let us remark that since $k = \partial_u$ is a Killing vector, it generates a symmetry of the (null) geodesic equation and is thus automatically a $(d+2)$ dimensional symmetry. In case $a$ is not constant then $ak$ is a generator linearly independent of $k$ and in that case the full $(d+2)$ dimensional algebra is an extension of the $(d+1)$ dimensional algebra by at least $ak$ and $k$. In this paper we will focus only on the extension by $ak$ only.

is constant, which is the case of our interest, so that (4.22) is an extension of the symmetries by the generator $k$. Now, observe that via (4.20) we have $[k, \hat{\xi}]_{d+2} = \lambda k$. This implies that $k$ does not commute with those symmetries, $\hat{\xi}$, that rescale the nonrelativistic Lagrangian and so when such symmetries are present, the extension provided by the Eisenhart lift is no longer central!

The whole discussion above is made explicit in the examples provided by the kinematical homogeneous spacetimes of galilean type that we have been studying in this paper. The Type I Newton–Cartan geometry (4.10, 4.11) describing the particle motion corresponds to the Bargmann structure

$$ds^2 = -2dudt - \alpha e^{\beta t}(x^a x^a)dt^2 + e^{\beta t}dx^a dx^a, \qquad k = \partial_u. \tag{4.23}$$

In particular, via the general formalism reviewed above, this implies that the null-geodesic equations of this lorentzian metric are equivalent to the equations of motion of the damped harmonic oscillator (3.4), as indeed can be verified explicitly. Furthermore, using (4.12) and (4.13), the vector fields (2.8), that form the kinematical algebra $\mathfrak{k}_{(\alpha,\beta)}$ and that are symmetries of the nonrelativistic system, get lifted to

$$\begin{aligned}
\hat{\xi}_{J_{ab}} &= \xi_{J_{ab}}, \\
\hat{\xi}_H &= \beta u \partial_u + \xi_H, \\
\hat{\xi}_{P_a} &= e^{\beta t}x^a \ddot{F}\partial_u + \xi_{P_a}, \\
\hat{\xi}_{B_a} &= e^{\beta t}x^a \dot{F}\partial_u + \xi_{B_a},
\end{aligned} \tag{4.24}$$

where $F$ was defined in equation (3.8). One verifies that these are indeed homothetic Killing vectors of the metric (4.23). Together with the Killing vector

$$\hat{\xi}_M = -\partial_u, \tag{4.25}$$

they form the Lie algebra

$$\begin{aligned}
[\hat{\xi}_{J_{ab}}, \hat{\xi}_{J_{cd}}] &= -\delta_{bc}\hat{\xi}_{J_{ad}} + \delta_{ac}\hat{\xi}_{J_{bd}} + \delta_{bd}\hat{\xi}_{J_{ac}} - \delta_{ad}\hat{\xi}_{J_{bc}}, \\
[\hat{\xi}_{J_{ab}}, \hat{\xi}_{B_c}] &= -\delta_{bc}\hat{\xi}_{B_a} + \delta_{ac}\hat{\xi}_{B_b}, \\
[\hat{\xi}_{J_{ab}}, \hat{\xi}_{P_c}] &= -\delta_{bc}\hat{\xi}_{P_a} + \delta_{ac}\hat{\xi}_{P_b}, \\
[\hat{\xi}_H, \hat{\xi}_{B_a}] &= \hat{\xi}_{P_a}, \\
[\hat{\xi}_H, \hat{\xi}_{P_a}] &= -\alpha\hat{\xi}_{B_a} - \beta\hat{\xi}_{P_a}, \\
[\hat{\xi}_{P_a}, \hat{\xi}_{B_b}] &= -\delta_{ab}\hat{\xi}_M, \\
[\hat{\xi}_H, \hat{\xi}_M] &= -\beta\hat{\xi}_M.
\end{aligned} \tag{4.26}$$

This algebra is (anti-)isomorphic to the one formed by the conserved charges (3.35) and provides the same extension of the kinematical algebra $\mathfrak{k}_{(\alpha,\beta)}$. In conclusion, we see that the Eisenhart lift provides a geometric realization, namely as homotheties of the metric (4.23), of this (non-centrally) extended algebra in a direct generalization of what happens for the usual free particle. The universality of this extension suggests to refer to it as the Bargmann extension of the galilean kinematical algebra $\mathfrak{k}_{(\alpha,\beta)}$, for any value of $\alpha$ and $\beta$.

## Acknowledgements

It is a pleasure to thank Andrew Beckett, Eric Bergshoeff, Mahmut Elbistan, Joaquim Gomis, Ross Grassie, Stefan Prohazka and Diederik Roest for useful discussions and conversations. CG is supported by the 2209-A program of TÜBİTAK under grant number 1919B012106282 and DVdB is partially supported by Boğaziçi University Research Fund under grant number 21BP2.

# A  Type I Newton–Cartan geometry: doublets, triplets and connections

Given a $(d+1)$ dimensional manifold $\mathcal{M}$, a Newton–Cartan[18] structure is a pair of tensors $(\tau_\mu, h^{\mu\nu})$, where $\tau_\mu$ everywhere spans the kernel of $h^{\mu\nu}$, which is symmetric and positive semi-definite. See e.g. [38] for a modern review.

For each Newton-Cartan structure one can define the following two additional notions:

- a *compatible NC-doublet* is an equivalence class $[(\hat{\tau}^\mu, \bar{h}_{\mu\nu})]$ of tensors $(\hat{\tau}^\mu, \bar{h}_{\mu\nu})$ such that

$$\tau_\mu \hat{\tau}^\nu + \bar{h}_{\mu\rho} h^{\rho\nu} = \delta_\mu^\nu, \tag{A.1}$$

  with the equivalence relation $(\hat{\tau}^\mu, \bar{h}_{\mu\nu}) \sim (\hat{\tau}'^\mu, \bar{h}'_{\mu\nu})$ if and only if

$$\hat{\tau}'^\mu = \hat{\tau}^\mu - h^{\mu\nu} \partial_\nu \Lambda, \tag{A.2}$$
$$\bar{h}'_{\mu\nu} = \bar{h}_{\mu\nu} + \tau_\mu \partial_\nu \Lambda + \tau_\nu \partial_\mu \Lambda, \tag{A.3}$$

- a *compatible NC-triplet* is an equivalence class $[(\tau^\mu, h_{\mu\nu}, C_\mu)]$ of tensors $(\tau^\mu, h_{\mu\nu}, C_\mu)$ such that

$$\tau_\mu \tau^\nu + h_{\mu\rho} h^{\rho\nu} = \delta_\mu^\nu, \qquad \tau^\mu \tau^\nu h_{\mu\nu} = 0, \tag{A.4}$$

  with the equivalence relation $(\tau^\mu, h_{\mu\nu}, C_\mu) \sim (\tau'^\mu, h'_{\mu\nu}, C'_\mu)$ if and only if

$$\tau_\mu \chi^\mu = 0, \tag{A.5}$$
$$\tau'^\mu = \tau^\mu - \chi^\mu, \tag{A.6}$$
$$h'_{\mu\nu} = h_{\mu\nu} + h_{\mu\rho} \chi^\rho \tau_\nu + h_{\nu\rho} \chi^\rho \tau_\mu + h_{\rho\sigma} \chi^\rho \chi^\sigma \tau_\mu \tau_\nu, \tag{A.7}$$
$$C'_\mu = C_\mu - h_{\mu\rho} \chi^\rho - \frac{1}{2} h_{\rho\sigma} \chi^\rho \chi^\sigma \tau_\mu + \partial_\mu \Lambda. \tag{A.8}$$

The notions of compatible NC-triplet and compatible NC-doublet are actually equivalent. This can be verified by considering the following explicit maps between them

$$[(\tau^\mu, h_{\mu\nu}, C_\mu)] \mapsto [(\hat{\tau}^\mu, \bar{h}_{\mu\nu})] = [(\tau^\mu - h^{\mu\nu} C_\nu, h_{\mu\nu} + \tau_\mu C_\nu + \tau_\nu C_\mu)], \tag{A.9}$$

$$[(\hat{\tau}^\mu, \bar{h}_{\mu\nu})] \mapsto [(\tau^\mu, h_{\mu\nu}, C_\mu)] = [(\hat{\tau}^\mu, \bar{h}_{\mu\nu} - \tau_\mu \tau_\nu \hat{\tau}^\rho \hat{\tau}^\sigma \bar{h}_{\rho\sigma}, -\frac{1}{2} \tau_\mu \hat{\tau}^\rho \hat{\tau}^\sigma \bar{h}_{\rho\sigma})]. \tag{A.10}$$

A Newton–Cartan structure together with a choice of compatible NC-triplet or -doublet was dubbed *Type I Newton–Cartan geometry* in [40], see also [30, 41]. As we review in Section 4.3 this geometric structure is also equivalent to a Bargmann structure in one dimension higher.

---

[18] Throughout the literature (slightly) different nomenclature has been used. As reviewed in [37] the pair $(\tau_\mu, h^{\mu\nu})$ are (the local components of characteristic tensor fields of) a $\mathrm{Gal}_0$-structure. Here $\mathrm{Gal}_0$ is the homogeneous Galilei group that can be defined as the subgroup of $\mathrm{GL}(d+1,\mathbb{R})$ consisting of the matrices $\begin{pmatrix} 1 & 0^T \\ v & A \end{pmatrix} \in \mathrm{GL}(d+1,\mathbb{R})$, with $v \in \mathbb{R}^d, A \in \mathrm{O}(d)$. Some authors for this reason speak of a Galilei (or galilean) structure, as in e.g. [10] and reserve the notion Newton(-Cartan) structure for a Galilei structure together with a (particular type of) compatible connection. In [38] a similar distinction is made by referring to a Galilei structure as a weak Newton-Cartan structure. In this paper we will however use the term Newton-Cartan structure interchangeably for Galilei, galilean or weak Newton-Cartan structure, i.e. to refer to the pair $(\tau_\mu, h^{\mu\nu})$, without any additional notions implied. Finally let us point out that in e.g. [39] the term leibnizian structure is used.

The first use of NC-doublets/triplets is that they define an affine connection compatible with the Newton–Cartan structure. If we assume the intrinsic torsion of the NC structure to vanish, i.e. $d\tau = 0$, then this connection takes the form

$$\Gamma^\rho_{\mu\nu} = \hat{\tau}^\lambda \partial_\mu \tau_\nu + \frac{1}{2} h^{\lambda\rho}(\partial_\mu \bar{h}_{\nu\rho} + \partial_\nu \bar{h}_{\mu\rho} - \partial_\rho \bar{h}_{\mu\nu}) \tag{A.11}$$

$$= \tau^\rho \partial_\mu \tau_\nu + \frac{1}{2} h^{\rho\sigma}(\partial_\mu h_{\nu\sigma} + \partial_\nu h_{\mu\sigma} - \partial_\sigma h_{\mu\nu}) - h^{\rho\sigma} K_{\sigma(\mu}\tau_{\nu)}, \tag{A.12}$$

where

$$K_{\mu\nu} = \partial_\mu C_\nu - \partial_\nu C_\mu. \tag{A.13}$$

One can directly verify that the connection above is invariant under the equivalences (A.2-A.3) and (A.6-A.8) and that (A.11) and (A.12) are related via (A.9). Essentially[19] all symmetric compatible connections are of the form (A.11, A.12), see e.g. [39, 42].

A second, related, application is that a choice of NC-doublet/triplet allows to write a Lagrangian for a particle on a manifold with Newton-Cartan structure:

$$S = \int \frac{1}{2} \frac{\bar{h}_{\mu\nu}\dot{x}^\mu \dot{x}^\nu}{\tau_\mu \dot{x}^\mu} d\sigma = \int \left( \frac{1}{2} \frac{h_{\mu\nu}\dot{x}^\mu \dot{x}^\nu}{\tau_\mu \dot{x}^\mu} + C_\mu \dot{x}^\mu \right) d\sigma. \tag{A.14}$$

Again it is straightforward to check that this action does not depend on a change of representative (A.2-A.3) and (A.6-A.8). The Euler–Lagrange equations of this action describe geodesic motion with respect to the affine connection (A.11, A.12), see (4.2).

## B  Symplectic Homotheties

Symmetries of the equations of motion are not necessarily symmetries of the action in the strict sense of leaving it invariant, rather they can also rescale the action with a constant. Using the term *homothety* for a diffeomorphism that multiplies a tensor with a constant we can rephrase the previous statement as saying that homotheties of the action map solutions of the equation of motion to other solutions. These symmetries in the more general sense have been considered in e.g., [43–48] (in a gravity context they are sometimes referred to as trombone symmetries [49]), and when considered on phase space they are generically *symplectic homotheties* (i.e. homotheties of the symplectic form) rather than symplectomorphisms.

Since results on symplectic homotheties appear somewhat scattered in the literature we collect some of the key concepts and their properties in this appendix. In addition, we will discuss the generalization of the extension of the Lie algebra of hamiltonian vector fields to the Poisson algebra of phase space functions to the homothetic case, where this extension is no longer central. This last result has not previously appeared in the literature as far as we are aware.

In this appendix $(\mathcal{S}, \Omega)$ will be assumed to be a symplectic manifold; that is $\Omega$ is a closed non-degenerate two-form on $\mathcal{S}$. One then defines a *homothetic symplectic vector field* as a vector field $X$ such that

$$\mathcal{L}_X \Omega = s\,\Omega, \quad s \in \mathbb{R}. \tag{B.1}$$

Given two such vector fields $X, Y$ one verifies $\mathcal{L}_{[X,Y]}\Omega = 0$ and thus the homothetic symplectic vector fields form a Lie algebra which we will denote as $\mathfrak{sym}(\mathcal{S})$. The definition defines a map $\sigma : X \mapsto s = \sigma(X)$ which is a Lie algebra homomorphism. In particular the kernel of this map, i.e. those vector fields $X$ for which $s = 0$, are the symplectic vector fields that form the Lie

---

[19]A subtle exception is provided by connections of the form A.12 where $K_{\mu\nu}$ is a two form that is not closed.

algebra $\mathfrak{sym}_0(\mathcal{S})$. Vector fields with $s \neq 0$ exist iff the symplectic form $\Omega$ is exact, i.e. $\Omega = d\theta$, since then $\Omega = \frac{1}{s} d(i_X \Omega)$. Note that this is the case for the canonical symplectic form on a cotangent bundle. It follows from the above that on an exact symplectic manifold $\mathcal{S}$ one has $\mathfrak{sym}(\mathcal{S})/\mathfrak{sym}_0(\mathcal{S}) = \mathbb{R}$.

Before we continue let us recall that for a symplectic vector field $X$ the one-form $i_X \Omega$ is closed, and that when $i_X \Omega$ is furthermore exact $X$ is called *hamiltonian*. Such vector fields form a subalgebra $\mathfrak{ham} \subset \mathfrak{sym}_0$. The non-degeneracy of the symplectic form allows to associate to every real function $f$ on $\mathcal{S}$ a hamiltonian vector field $X_f$ via

$$i_{X_f} \Omega = df \, . \tag{B.2}$$

The map $f \mapsto X_f$ is a Lie algebra (anti-)homomorphism if we equip the space of functions with the Poisson bracket $\{f, g\} = \Omega(X_f, X_g)$, i.e.,

$$[X_f, X_g] = X_{-\{f,g\}} \, . \tag{B.3}$$

Since the kernel of the map $f \mapsto X_f$ are the constant functions, which are central with respect to the Poisson bracket, one observes the well known fact that the Poisson algebra of functions is a central extension of the Lie algebra of hamiltonian vector fields.

Let us now get back to the homothetic generalization. Assuming from now on that $\Omega = d\theta$ is exact we can define an[20] *Euler vector field $E$* via[21]

$$i_E \Omega = \theta \, . \tag{B.4}$$

Now remark that $X - \sigma(X)E$ is symplectic when $X$ is homothetic symplectic. If furthermore $X - \sigma(X)E$ is hamiltonian we say that $X$ is *homothetic hamiltonian* (with respect to $\theta$). It follows directly from this definition that for every such vector field there exists a function $f$ and a real number $s$ such that $X = X_{(s,f)}$, where

$$X_{(s,f)} = sE + X_f \, . \tag{B.5}$$

A short computation reveals that

$$[X_{(s_1,f_1)}, X_{(s_2,f_2)}] = X_{-[[(s_1,f_1),(s_2,f_2)]]} \, , \tag{B.6}$$

where

$$[[(s_1,f_2),(s_2,f_2)]] = (0, \{f_1,f_2\} + s_1(f_2 - E[f_2]) - s_2(f_1 - E[f_1])) \, . \tag{B.7}$$

First of all this implies that the homothetic hamiltonian vector fields form a Lie algebra, that we denote as[22] $\mathfrak{ham}_\theta(\mathcal{S})$. It also makes clear that $\mathfrak{ham}(\mathcal{S})$ is an ideal in $\mathfrak{ham}_\theta(\mathcal{S})$. Indeed, one has the Lie algebra extension by derivation $0 \to \mathfrak{ham}(\mathcal{S}) \to \mathfrak{ham}_\theta(\mathcal{S}) \to \mathbb{R} \to 0$ [44]. Furthermore $[[\cdot, \cdot]]$ defines a Lie[23] algebra on the space $C_\theta^\infty(\mathcal{S}) = \mathbb{R} \oplus C^\infty(\mathcal{S})$. By construction the map $(s, f) \mapsto X_{(s,f)}$ is a Lie algebra homomorphism, and via (B.5) one infers the kernel is given by $(0, c)$ with $c \in \mathbb{R} \subset C^\infty(\mathcal{S})$ being a constant function. So also in the homothetic generalization it remains true that $C_\theta^\infty(\mathcal{S})$ is a one-dimensional extension of $\mathfrak{ham}_\theta(\mathcal{S})$, but it is no longer true that this extension is central, since

$$[[(0,c),(s,f)]] = (0, -sc) \, . \tag{B.8}$$

---

[20]Note that the choice of the symplectic potential $\theta$, and hence the associated Euler vector field, is not unique. The possible choices differ by an arbitrary closed form. So although one could choose to write $E_\theta$ to emphasize the dependence on the choice of $\theta$ we will refrain from doing so to ease notation.

[21]Since $\Omega$ is non-degenerate $E$ is unique given $\theta$.

[22]Indeed different choices of symplectic potential $\theta$ lead to different algebras $\mathfrak{ham}_\theta(\mathcal{S})$, these are isomorphic iff the two potentials are cohomologous.

[23]Note that although $[[\cdot, \cdot]]$ is a Lie bracket, it is neither a Poisson nor Jacobi bracket.

A summary of the various Lie algebra extensions is provided below:

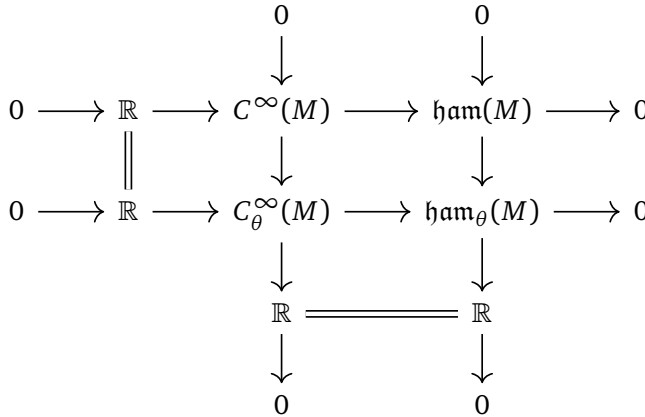

All arrows in the above commutative diagram are Lie algebra homomorphisms. The four Lie algebra extensions discussed in the text correspond to the two vertical and two horizontal short exact sequences.

In a physical context, the homothetic symplectic vector fields are only relevant when, in addition, they leave invariant Hamilton's equations. This extra condition can alternatively be expressed by considering the hamiltonian action

$$S[\gamma] = -\int_{t_i}^{t_f} (\gamma^* \theta + \gamma^* h \, dt), \tag{B.9}$$

with $\gamma$ a curve and $h$, the Hamiltonian, a function on $\mathcal{S}$, and requiring that under $\delta\gamma = X_{(s,f)}$

$$\delta S = \lambda S + q(\gamma_i, \gamma_f), \tag{B.10}$$

where $q$ is some arbitrary function of the endpoints of the curve.

Define now (the generator of) a *symmetry* of an exact hamiltonian system $(\mathcal{S}, \theta, h)$ to be a homothetic hamiltonian vector field $X_{(s,f)}$ such that[24]

$$\mathcal{L}_{X_{(s,f)}} h = s h + \partial_t f . \tag{B.11}$$

It then follows that under such symmetries indeed (B.10) is guaranteed. To verify this, remark that

$$\delta \int_{t_i}^{t_f} \gamma^* h \, dt = \int_{t_i}^{t_f} \gamma^* (\mathcal{L}_{X_{(s,f)}} h) \, dt , \tag{B.12}$$

while

$$\delta \int_{t_i}^{t_f} \gamma^* \theta = s \int_{t_i}^{t_f} \gamma^* \theta + (f - \mathcal{L}_E f)|_{\gamma_i}^{\gamma_f} - \int_{t_i}^{t_f} \gamma^* \partial_t f \, dt . \tag{B.13}$$

Here we used that[25]

$$\delta(\gamma^* \theta) = \gamma^* \mathcal{L}_{X_{(s,f)}} \theta - \gamma^* (\partial_t \mathcal{L}_E f) \quad \text{and} \quad \mathcal{L}_{X_{(s,f)}} \theta = s \theta + d(f - \mathcal{L}_E f) . \tag{B.14}$$

---

[24]Note that (B.11) can equivalently be rewritten as $\mathcal{L}_{X_{(s,f)}} X_h + X_{\partial_t f} = 0$ or $[[(0,h),(s,f)]] = (0, \partial_t f)$.

[25]Remark: $i_{X_f} \theta = i_{X_f} i_E \Omega = -i_E df = -\mathcal{L}_E f$ and $i_E \theta = i_E i_E \Omega = 0$.

Now observe

$$\frac{d}{dt}f \;=\; \mathcal{L}_{X_h}f + \partial_t f \tag{B.15}$$

$$=\; -\mathcal{L}_{X_{(s,f)}}h + s\mathcal{L}_E h + \partial_t f \tag{B.16}$$

$$=\; s(\mathcal{L}_E h - h). \tag{B.17}$$

We thus see that a symmetry $X_{(s,f)}$ leads to a conserved charge $f$ when

$$s = 0, \quad \text{or} \quad \mathcal{L}_E h = h. \tag{B.18}$$

We should point out that the condition (B.18) depends on the choice of Euler vector field (and thus choice of symplectic potential $\theta$).

In summary, for the existence of a conserved charge $f$ it is thus sufficient that there exists an Euler vector field for which both (B.18) and (B.11) hold.

## C  Low dimensional cases

When the number of spatial dimension is two or less, $d \leq 2$, there appear some exceptions to the discussion of $d > 2$ in the main text. When $d = 2$ there exists an additional kinematical algebra of galilean type [3], and both for $d = 1$ and $d = 2$ the invariant connections are less constrained than in higher dimensions. In this appendix we shortly go over the various cases and point out the similarities and subtle differences with the generic case discussed in the main text.

### C.1  $d = 2$

The case of two spatial dimensions is special, since it is the unique dimension where there is a second rotationally invariant 2-tensor; apart from the generic $\delta_{ab}$ one now also has $\epsilon_{ab}$. This extra tensor can appear in the symmetry algebra, as well as in the invariant connection.

### C.1.1  $\mathcal{M}_{(\alpha,\beta)}$

First we discuss the standard kinematical homogeneous spaces of galilean type $\mathcal{M}_{(\alpha,\beta)}$, i.e. those based on the algebra $\mathfrak{k}_{(\alpha,\beta)}$, defined in (2.1) and (2.2). The homogeneous space itself is constructed as in the higher dimensional cases in main text. A subtle difference is in the classification of invariant affine connections on this homogeneous space. A short calculation reveals them to take the following form in modified exponential coordinates

$$\begin{aligned}
\Gamma^t_{tt} &= \kappa + \iota\,, \\
\Gamma^a_{tb} &= \delta^a_b \kappa + \kappa' \epsilon_{ba}\,, \\
\Gamma^a_{bt} &= \delta^a_b(\beta + \iota) - \kappa' \epsilon_{ba}\,, \\
\Gamma^a_{tt} &= \alpha x^a\,.
\end{aligned} \tag{C.1}$$

Comparing to (2.14) one sees that there is one additional free parameter $\kappa'$. However, just like the other two Nomizu parameters $\kappa$ and $\iota$ it drops out of the autoparallel equation, which upon fixing the parameter as $\sigma = t$, takes the form

$$\ddot{x}^a + \beta \dot{x}^a + \alpha x^a = 0\,. \tag{C.2}$$

This is exactly the same damped oscillation equation as in higher dimensions and so the analysis of the symmetries and conserved charges is identical to that of the main text.

### C.1.2  $\tilde{m}_{(\gamma,\chi)}$

The existence of $\epsilon_{ab}$ leads to an additional class of kinematical algebras of galilean type in two spatial dimensions. The additional homogeneous kinematical spacetimes are called $S12_{\gamma,\chi}$ in [3], but we will refer to them as $\tilde{m}_{(\gamma,\chi)}$. Their underlying kinematical Lie algebra $\tilde{\mathfrak{k}}_{(\gamma,\chi)}$ shares the brackets (2.1) with the generic algebra $\mathfrak{k}_{(\gamma,\chi)}$, but the non-vanishing brackets in (2.2) get replaced by

$$[H,B_a] = -P_a, \qquad [H,P_a] = \gamma B_a + (1+\gamma)P_a - \chi\epsilon_{ab}(P_b + B_b). \tag{C.3}$$

The associated homogeneous space is then similarly defined as $\tilde{m}_{(\gamma,\chi)} = \tilde{\mathcal{K}}_{(\gamma,\chi)}/\mathcal{H}$. As in the generic case one defines modified exponential coordinates (2.5). In these coordinates one can compute the vector fields generating the $\tilde{\mathcal{K}}_{(\gamma,\chi)}$ action:

$$
\begin{aligned}
\xi_{J_{ab}} &= x^b \partial_a - x^a \partial b, \\
\xi_H &= \partial_t, \\
\xi_{B_a} &= A(t)\partial_a + B(t)\epsilon_{ab}\partial_b, \\
\xi_{P_a} &= \dot{A}(t)\partial_a + \dot{B}(t)\epsilon_{ab}\partial_b,
\end{aligned}
\tag{C.4}
$$

where now

$$
\begin{aligned}
A(t) &= \frac{e^{-t}(\gamma-1) + e^{-t\gamma}(\chi\sin(t\chi) - (\gamma-1)\cos(t\chi))}{(\gamma-1)^2 + \chi^2}, \\
B(t) &= \frac{e^{-t}\chi + e^{-t\gamma}(-\chi\cos(t\chi) - (\gamma-1)\sin(t\chi))}{(\gamma-1)^2 + \chi^2}.
\end{aligned}
\tag{C.5}
$$

The non-zero components of an invariant affine connection can then be computed to be of the form

$$
\begin{aligned}
\Gamma^t_{tt} &= (\kappa + \iota), \\
\Gamma^a_{tb} &= (\kappa\delta^a_b + \kappa'\epsilon_{ba}), \\
\Gamma^a_{bt} &= (1+\gamma)\delta^a_b + \chi\epsilon_{ab} + (\iota\delta^a_b - \kappa'\epsilon_{ba}), \\
\Gamma^a_{tt} &= \gamma x^a + \chi\epsilon_{ab}x^b,
\end{aligned}
\tag{C.6}
$$

where $\kappa, \kappa', \iota$ are three unconstrained, real constants.

Upon fixing the parameter $\sigma = t$ the autoparallel equation associated to this invariant connection reads

$$\ddot{x}^a + \dot{x}^a + \gamma(x^a + \dot{x}^a) + \chi\epsilon_{ab}(x^b + \dot{x}^b) = 0. \tag{C.7}$$

Note that these equations have a rather different[26] structure than that of the damped harmonic oscillator (3.4).

Somewhat surprisingly there exists a one-parameter family of Lagrangians (not related by a total derivative term) with (C.7) as their Euler–Lagrange equations:

$$L_\theta = \frac{e^{(1+\gamma)t}}{2}M_{ab}(\theta)(\dot{x}^a\dot{x}^b - A_{bc}x^a x^c), \tag{C.8}$$

---

[26]Formally the equations (C.7) are some complexification of the equations (3.4), since by introducing $z = x^1 + ix^2$ and $\alpha = \gamma - i\chi, \beta = 1 + \gamma - i\chi$ the equations (C.7) take the form $\ddot{z} + \beta\dot{z} + \alpha z = 0$. Indeed also the the algebra $\tilde{\mathfrak{k}}_{(\gamma,\chi)}$ can formally be brought into the form $\mathfrak{k}_{(\alpha,\beta)}$ via a complexification of boost and translation generators. I.e. defining $\mathbb{B} = B_1 - iB_2$, $\mathbb{P} = P_1 - iP_2$ one finds $[H,\mathbb{B}] = -\mathbb{P}$ and $[H,\mathbb{P}] = \alpha\mathbb{B} + \beta\mathbb{P}$.

where

$$(M_{ab}) = m \begin{pmatrix} \cos(\chi t + \theta) & \sin(\chi t + \theta) \\ \sin(\chi t + \theta) & -\cos(\chi t + \theta) \end{pmatrix}, \quad (A_{ab}) = \begin{pmatrix} \gamma & \chi \\ -\chi & \gamma \end{pmatrix}. \tag{C.9}$$

One easily verifies that the actions $S_\theta = \int L_\theta dt$ are invariant under the transformations generated by $\xi_{J_{ab}}$, $\xi_{B_a}$ and $\xi_{P_a}$ listed in (C.4) and these thus constitute symmetries in the usual sense. Under a time translation one however finds

$$\delta_H S_\theta = (1 + \gamma) S_\theta + \chi S_{\theta + \frac{\pi}{2}}. \tag{C.10}$$

In the first term we recognize a homothetic rescaling of the action, as we encountered in the main text. The second term mixes the action $S_\theta$ with another action $S_{\theta + \frac{\pi}{2}}$. So time translations are no longer a symmetry in the standard sense. But since $S_\theta$ and $S_{\theta + \frac{\pi}{2}}$ both share the same Euler–Lagrange equations, it follows that indeed the time-translations leave these Euler–Lagrange equations invariant – as can be directly verified from the form (C.7) – and so (C.10) still is a symmetry in a more general sense.

One could repeat the Hamiltonian analysis and try to find a way to represent the symmetry algebra in terms of conserved charges. It is rather straightforward to compute canonical momenta $p_a$ and their transformation under the symmetries. One can then verify that indeed the corresponding phase space space vector fields reproduce the Lie algebra $\tilde{\mathfrak{k}}_{(\gamma,\chi)}$. What complicates a further analysis however is that the vector field corresponding to time translations is no longer symplectic, and not even homothetic symplectic. Indeed, it mixes the canonical symplectic form with a non-canonical one. It would be interesting to understand this more general notion of symplectic transformation, construct an analog of hamiltonian vector fields and a corresponding generalized notion of Poisson bracket. I.e. extending the discussion of Appendix B to these more general transformations. This would however take us too far from the main topic of this paper and so we leave this as an interesting open problem.

## C.2 $d = 1$

The case of one spatial dimension is special, since there are no rotations in this case. This does not change anything from the point of view of the kinematical algebras or homogeneous spaces, which remain only of the type $\mathcal{M}_{(\alpha,\beta)}$. In particular the vector fields generating the group action remain the same, i.e., (2.8). A small but not completely trivial change is the presence of an extra freedom in the invariant connections, which take the form

$$\begin{aligned} \Gamma^t_{tt} &= (\kappa + \iota), \\ \Gamma^1_{t1} &= \kappa, \\ \Gamma^1_{1t} &= (\beta + \iota), \\ \Gamma^1_{tt} &= \alpha x - \psi. \end{aligned} \tag{C.11}$$

Comparing to the case in generic dimensions, (2.14) one sees there is an additional free parameter $\psi$.

Unlike in all other cases, the Nomizu parameter $\psi$, particular to $d = 1$, does not drop out of the autoparallel equation:

$$\ddot{x} + \beta \dot{x} + \alpha x = \psi. \tag{C.12}$$

The extra Nomizu parameter corresponds physically speaking to an extra constant force $m\psi$.

We should point out that when $\alpha \neq 0$ adding $\psi$ is rather trivial, since one can obtain all expressions from the $\psi = 0$ case simply by the replacement $x \to x - \alpha^{-1}\psi$. This is true for

the action (3.5), and the conserved charges (3.21-3.23), which thus take the form

$$P = \dot{F}p - me^{\beta t}\ddot{F}(x - \alpha^{-1}\psi),$$
$$B = Fp - me^{\beta t}\dot{F}(x - \alpha^{-1}\psi). \tag{C.13}$$

Also the Euler vector field (3.28) gets shifted,

$$\tilde{E} = \frac{1}{2}(x - \alpha^{-1}\psi)\partial_x + \frac{p}{2}\partial_p \tag{C.14}$$

and one should consider the canonical Hamiltonian

$$\tilde{h} = h + m\frac{\psi^2}{2\alpha}e^{\beta t} = \frac{e^{-\beta t}}{2m}p^2 + \frac{m\alpha e^{\beta t}}{2}(x - \alpha^{-1}\psi)^2. \tag{C.15}$$

This to guarantee one has the crucial relationship $\mathcal{L}_{\tilde{E}}\tilde{h} = \tilde{h}$, see Appendix B. One additionally verifies that the vector field generating the phase space time translation takes the form

$$\Xi_H = X_{(\beta,H)} = \beta\tilde{E} + X_H, \qquad H = -(\tilde{h} + \frac{\beta}{2}(x - \alpha^{-1}\psi)p). \tag{C.16}$$

The realization of the kinematical algebra $\mathcal{K}_{(\alpha,\beta)}$ in terms of conserved charges then goes through as in the main text.

Somewhat to our surprise, the case $\alpha = 0$ is subtly different. One still has an action for (C.12), which in this case reads

$$S = \int \frac{me^{\beta t}}{2}(\dot{x}^2 + 2\psi x)\,dt. \tag{C.17}$$

For the spatial translation and boost we find satisfying and rather standard results. There exist the conserved charges[27]

$$P = e^{-\beta t}p + m(\beta x - \psi t),$$
$$B = \beta^{-1}(1 - e^{-\beta t})p - mx + m\psi\beta^{-2}(1 + \beta t - e^{\beta t}), \tag{C.18}$$

and the phase space transformations are generated by the corresponding hamiltonian vector fields $X_P$ and $X_B$ respectively. But when $\beta \neq 0$, there does not exist an Euler field and charge $H$ for which $\Xi_H = \beta E + X_H$ while at the same time also the conditions (3.31) are met. This implies we cannot represent the algebra $\mathfrak{k}_{(0,\beta)}$ in terms of conserved charges when $\beta \neq 0$ and $\psi \neq 0$.

The special case when both $\alpha = \beta = 0$ does have a (simple) solution. In this case the time translation vector field is hamiltonian: $\Xi_H = X_H$ with

$$H = -h = -\frac{p^2}{2m} + m\psi x. \tag{C.19}$$

Since $h$ is time independent this charge is indeed conserved. In addition one can set $\beta = 0$ in (C.18) to get conserved charges

$$P|_{\alpha=\beta=0} = p - \psi mt,$$
$$B|_{\alpha=\beta=0} = pt - mx - \psi m\frac{t^2}{2}. \tag{C.20}$$

---

[27]Note that both expressions in (C.13) are singular in the $\alpha \to 0$ limit. This singularity can be removed by adding to the expressions for $P$ and $B$ the constants $\alpha^{-1}\beta m\psi$ and $-\alpha^{-1}m\psi$, respectively. The charges in (C.18) are then the $\alpha \to 0$ limit of these 'regularized' charges.

We thus uncover the curiosity that in one dimension a free particle, i.e., $\alpha = \beta = 0$, remains Galilei-invariant even in the presence of a constant force $m\psi$. In other words, the fact that in $d > 1$ the presence of a constant force breaks this invariance is only due to the fact that it breaks rotational invariance, which is not an issue when $d = 1$. Note however that the Bargmann central extension appears slightly differently when $\psi \neq 0$:

$$\{P,B\} = m, \qquad \{H,B\} = -P, \qquad \{H,P\} = \psi m. \tag{C.21}$$

It can be put into a standard form by defining $\tilde{H} = H + \psi B = -\frac{(p-m\psi t)^2}{2}$:

$$\{P,B\} = m, \qquad \{\tilde{H},B\} = -P, \qquad \{\tilde{H},P\} = 0. \tag{C.22}$$

Actually this redefinition of time translations is an isomorphism of the 1d Galilei algebra $\mathfrak{k}_{(0,0)}$.

## D Conformally equivalent Eisenhart lifts

In Section 4 we reviewed how mechanical motion can be described in a covariant fashion using Newton-Cartan geometry and how that description in turn is equivalent to null geodesic motion with respect to a particular Lorentzian metric. In this appendix we shortly recall that these descriptions are unique only up to a conformal redefinition. We refer to [34] for a complete discussion, here we simply mention how some key formulae of Section 4 behave under such a conformal redefinition and then focus on the case of interest in this paper, namely the damped harmonic motion describing a free particle on $\mathcal{M}_{(\alpha,\beta)}$, and how some results in this paper are related to those in [15].

The starting observation is that the Type I NC geometries $(\tau_\mu, h^{\mu\nu}; \hat{\tau}^\mu, \bar{h}_{\mu\nu})$ and $(\tau_\mu^c, h_c^{\mu\nu}; \hat{\tau}_c^\mu, \bar{h}_{\mu\nu}^c)$ lead to the same action (4.1) if they are conformally related as

$$\tau_\mu^c = e^\psi \tau_\mu, \quad h_c^{\mu\nu} = e^{-\psi} h^{\mu\nu}, \quad \hat{\tau}_c^\mu = e^{-\psi} \hat{\tau}^\mu, \quad \bar{h}_{\mu\nu}^c = e^\psi \bar{h}_{\mu\nu}, \tag{D.1}$$

for an arbitrary function $\psi$. So really the particle action (4.1) only depends on a conformal class of type I NC geometries.

Under such a conformal redefinition of the NC geometry essentially all results of Section 4 remain valid, but some of the parameters will depend on the choice of conformal prefactor. For example, the parameters characterizing the invariance of the action under symmetries, as in (4.7), transform as

$$\zeta_c = \zeta + \mathcal{L}_\xi \psi, \qquad \lambda_c = \lambda, \qquad K_c = K. \tag{D.2}$$

Since the lift (4.20) of the vector fields generating the symmetries only depends on $\lambda$ and $K$ it remains invariant under a conformal redefinition. In turn this implies that the lifted symmetry algebra is independent of the choice of conformal prefactor $\psi$. Of course this is not the case for the Lorentzian metric (4.14): unsurprisingly, it transforms by a conformal rescaling

$$g_{AB}^c = e^\psi g_{AB}. \tag{D.3}$$

Note that since $\psi$ is, by construction, independent of $u$, the vector field $k = \partial_u$ remains null and Killing for all choices of conformal factor.

The transformations (D.2) and (D.3) are compatible in that they leave (4.21) invariant i.e., one also has that

$$\mathcal{L}_{\hat{\xi}} g_{AB}^c = (\lambda_c + \zeta_c) g_{AB}^c, \tag{D.4}$$

so that for any choice of conformal factor the lifted symmetries will be conformal Killing vectors. Let us remark that since $\zeta$ changes under a change of conformal factor, see (D.2), one

may be able to find special conformal factors for which all $\lambda_c + \zeta_c$ are constant and the lifted symmetries are homotheties, or even such that $\lambda_c + \zeta_c = 0$ for all vector fields so that the lifted symmetries become isometries. This turns out to be possible for the geometries of interest.

Let us now specialize to motion on $\mathcal{M}_{(\alpha,\beta)}$, the case of interest in this paper. Allowing for a generic conformal factor, the NC geometry (4.10, 4.11) becomes

$$
\begin{aligned}
\tau_\mu^c &= e^\psi \delta_\mu^t, \\
h_c^{\mu\nu} &= e^{-\beta t - \psi} \delta_a^\mu \delta_a^\nu, \\
\hat{\tau}_c^\mu &= e^{-\psi} \delta_t^\mu, \\
\bar{h}_{\mu\nu}^c &= e^{\beta t + \psi} \delta_\mu^a \delta_\nu^a - \alpha e^{\beta t + \psi} x^a x^a \delta_\mu^t \delta_\nu^t.
\end{aligned}
\tag{D.5}
$$

The Lorentzian metric describing the Eisenhart lift (4.23) then gains an overall conformal prefactor:

$$
ds_c^2 = e^\psi \left( -2 du\, dt - \alpha e^{\beta t} x^a x^a dt^2 + e^{\beta t} dx^a dx^a \right), \qquad k = \partial_u.
\tag{D.6}
$$

To make the equivalence to [15] explicit, make the coordinate transformation

$$
u' = u + \Lambda, \qquad \Lambda = \frac{\beta}{2} e^{\beta t} x^a x^a.
\tag{D.7}
$$

Note that this is a special type of coordinate transformation corresponding to the redefinitions of the NC doublet, as mentioned in (4.16). The Bargmann structure (D.6) then becomes

$$
ds_c^2 = e^\psi \left( -(2 du' + \alpha e^{\beta t} x^a x^a dt) dt + e^{\beta t} (dx^a + \beta x^a dt)(dx^a + \beta x^a dt) \right), \quad k = \partial_{u'}.
\tag{D.8}
$$

Upon the choice $\psi = -\beta t$ this becomes exactly the metric described in Appendix A of [15]. In conclusion, while the lifted vector fields (4.24), that form the algebra (4.26), generate homotheties of the metric (4.23), these same transformations are isometries of (D.8) when $\psi = -\beta t$. This follows from (D.4) and the fact that $\zeta_H^c = -\beta = -\lambda_H^c$, while $\zeta_H = 0, \lambda_H = \beta$.

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
