# Peer review of "Particle dynamics on torsional galilean spacetimes"

_SciPost Physics, doi:SciPost Phys. 14, 059 (2023)_

## Round 1 · Referee Report · Anonymous (Referee 1) · 2022-11-28

Strengths

1- Extremely well structured and easy to follow 2- Self-contained 3- Very interesting physics supported by solid mathematics

Weaknesses

None that were apparent to me.

Report

This work focuses on torsional Galilean spacetimes and free particle motion on these backgrounds. The authors show with great care that in suitable coordinates, the equations of motion reduce to those of a damped harmonic oscillator. A good portion of this work is dedicated to setting the stage for this main result. The authors first provide a review of torsional Galilean spacetimes and introduce the necessary mathematics to describe these kinematical spacetimes before they derive their main result. The remainder of this work is dedicated to a detailed discussion on how to realize the underlying kinematical Lie algebra as symmetries of the damped harmonic oscillator (including a covariant description) and in terms of conserved charges.

This paper is exceptionally well written and a joy to read. All necessary concepts and underlying mathematical structures are appropriately introduced, and the relevant literature is adequately referred to. The authors also take great care to provide a physical interpretation of the (maybe for some readers abstract) mathematical notions that underlie their analysis.

Maybe the only "weakness" (though this depends on personal preferences) is that this work derives a fascinating result -- the appearance of the damped harmonic oscillator as the equations of motion -- and does not follow up on it besides some hints on future directions. Something as fundamental as the (damped) harmonic oscillator might hint at a lot of interesting physics and relations to other fields to be explored.

In summary, this is excellent work with interesting physical interpretations written clearly and concisely, and I recommend publication in its current form.

Requested changes

None.

---

## Round 1 · Referee Report · Anonymous (Referee 2) · 2022-12-2

Strengths

1- Contains novel and interesting results.
2- Use powerful mathematical methods.
2- It is very rigorous and clearly written.

Weaknesses

- I do not see any weak point.

Report

The dynamics of free particles moving in geodesics on a torsional galilean spacetime was analyzed. One of the main results of the article is that the dynamics of free particles, using a specific coordinate system, is described by a damped harmonic oscillator. Damping is then related to the presence of torsion, generalizing previous results found in the literature.

The action principle, used to describe the dynamics of the particles, is not invariant under time translations, but it rescales by a constant. As a consequence, the equations of motion are invariant under this symmetry, and an associated conserved charge can be constructed. The kinematical algebra is then realized using a suitable modification of the Poisson brackets. Surprisingly, if the standard Poisson brackets are used, the charges also close in an algebra.

Finally, in section 4 the dynamics is described in covariant way using Newton-Cartan geometry, without specifying a particular coordinate system.

The article is clear, well-written and rigorous. The results are novel and very interesting. In my opinion this is really an excellent paper.

I recommend the article for publication in its current form.

Requested changes

1- No changes

---

## Editorial Decision

published